# Excluding the Irrelevant: Focusing Reinforcement Learning through Continuous Action Masking

**Roland Stolz**[1,*]               **Hanna Krasowski**[1,2,*]

**Jakob Thumm**[1]     **Michael Eichelbeck**[1]     **Philipp Gassert**[1,3]     **Matthias Althoff**[1]

[1]Technical University of Munich, [2]University of California, Berkeley,
[3]Munich Center for Machine Learning
{roland.stolz, hanna.krasowski}@tum.de

## Abstract

Continuous action spaces in reinforcement learning (RL) are commonly defined as multidimensional intervals. While intervals usually reflect the action boundaries for tasks well, they can be challenging for learning because the typically large global action space leads to frequent exploration of irrelevant actions. Yet, little task knowledge can be sufficient to identify significantly smaller state-specific sets of relevant actions. Focusing learning on these relevant actions can significantly improve training efficiency and effectiveness. In this paper, we propose to focus learning on the set of relevant actions and introduce three continuous action masking methods for exactly mapping the action space to the state-dependent set of relevant actions. Thus, our methods ensure that only relevant actions are executed, enhancing the predictability of the RL agent and enabling its use in safety-critical applications. We further derive the implications of the proposed methods on the policy gradient. Using proximal policy optimization (PPO), we evaluate our methods on four control tasks, where the relevant action set is computed based on the system dynamics and a relevant state set. Our experiments show that the three action masking methods achieve higher final rewards and converge faster than the baseline without action masking.

## 1   Introduction

Reinforcement learning (RL) can solve complex tasks in areas such as robotics [13], games [35], and large language models [26]. Yet, training RL agents is often sample-inefficient due to frequent exploration of actions, which are irrelevant to learning a good policy. Irrelevant actions are actions that are either physically impossible, forbidden due to some formal specification, or evidently counterproductive for solving the task. Since the global action space is typically large in relation to the relevant actions in each state, exploring these actions frequently can introduce unnecessary costs, lead to slow convergence, or even prevent the agent from learning a suitable policy.

Action masking mitigates this problem by constraining the exploration to the set of relevant state-specific actions, which can be obtained based on task knowledge. For example, when there is no opponent within reach in video games, attack actions are masked from the action space [43]. Leveraging task knowledge through action masking usually leads to faster convergence and also improves the predictability of the RL agent, especially when the set of relevant actions has a specific

---

*The first two authors contributed equally to this work.

38th Conference on Neural Information Processing Systems (NeurIPS 2024).

notion, such as being the set of safe actions. For instance, if the set of relevant actions is a set of verified safe actions, action masking can be used to provide safety guarantees [10, 19].

When the relevant action set is easy to compute, action masking usually benefits RL by improving the sample efficiency and is the quasi-standard for discrete action spaces [34, 46], e.g., in motion planning [8, 18, 23, 30, 42], games [14, 15, 43], and power systems [21, 38]. However, real-world systems operate in continuous space and discretizing it might prevent learning optimal policies. Furthermore, simulation of real-world systems is computationally expensive, and developing RL agents for them often requires additional real-world training [45]. Thus, sample efficiency is particularly valuable for these applications.

In this work, we propose three action masking methods for continuous action spaces. They can employ convex set representations, e.g., polytopes or zonotopes, for the relevant action set. Our action masking methods generalize previous work in [19], which is constrained to intervals as relevant action sets. This extends the applicability of continuous action masking to expressive convex relevant action sets, which is especially useful when action dimensions are coupled, e.g., for a thrust-controlled quadrotor. To integrate the relevant action set into RL, we introduce three methods: the *generator mask*, which exploits the generator representation of a zonotope, the *ray mask*, which projects an action into the relevant action set based on radial directions, and the *distributional mask*, which truncates the policy distribution to the relevant action set. In summary, our main contributions are:

- We introduce continuous action masking based on convex sets representing the state-dependent relevant action sets;
- We present three methods to utilize the relevant action sets and derive their integration in the backward pass of RL with stochastic policies;
- We evaluate our approach on four benchmark environments that demonstrate the applicability of our continuous action masking approaches.

## 2   Related literature

Action masking has been mainly applied to discrete action spaces [8, 10, 14, 15, 19, 18, 21, 23, 30, 38, 41, 42, 43, 46]. Huang et al. [15] derive implications on policy gradient RL and evaluate their theoretical findings on real-time strategy games. They show that masking actions leads to higher training efficiency and scales better with an increasing number of actions than penalizing the agent for selecting irrelevant actions. Huo et al. [14] have extended [15] to off-policy RL and have observed similar empirical results.

Action masking for discrete action spaces can be categorized by the purpose of the state-dependent relevant action set. Often, the set is obtained by removing irrelevant actions based on task knowledge [8, 14, 15, 21, 30, 42], e.g., executing a harvesting action before goods are produced [15]. While the relevant action sets are usually manually engineered, a recent study [46] takes a data-driven approach and identifies redundant actions based on similarity metrics. Another common interpretation of the relevant action set is that it only includes safe actions [10, 18, 23, 38, 41]. These works typically use the system dynamics to verify the safety of actions. A safe action avoids defined unsafe areas or complies with logic formulas. In our experiments, the relevant action sets are either state-dependent safe action sets or a global relevant action set modeling power supply constraints.

For continuous action spaces, there is work on utilizing action masking with multidimensional intervals (hereafter only referred to as intervals) as relevant action sets [19]. In particular, the proposed continuous action masking represents relevant action sets by intervals that reflect safety constraints and employs straightforward re-normalization to map the action space to the relevant action set. In this paper, we generalize to more expressive relevant action sets and demonstrate the applicability to four benchmark environments.

## 3   Preliminaries

As the basis for our derivations of the three masking methods, we provide a concise overview of RL with policy gradients. Further, we define the considered system dynamics as well as the set representations used in this work.

## 3.1 Reinforcement learning with policy gradients

A Markov decision process is a tuple $(\mathcal{S}, \mathcal{A}, T, r, \gamma)$ consisting of the following elements: the observable and continuous state set $\mathcal{S} \subset \mathbb{R}^{n^{\mathcal{S}}}$, the action set $\mathcal{A} \subset \mathbb{R}^N$, the state-transition distribution $T(s'|a, s)$, which is stationary and describes the transition probability to the next state $s' \in \mathcal{S} \subset \mathbb{R}^{n^{\mathcal{S}}}$ from the current state $s \in \mathcal{S}$ when executing action $a \in \mathcal{A}$, the reward $r : \mathcal{S} \times \mathcal{A} \to \mathbb{R}$, and the discount factor $\gamma$ for future rewards [37]. The goal of RL is to learn a parameterized policy $\pi_\theta(a|s)$ that maximizes the expected reward $\max_\theta \mathbb{E}_{\pi_\theta} \sum_{t=0}^{\infty} \gamma^t r(s_t, a_t)$.

For policy gradient algorithms, learning the optimal policy $\pi_\theta^*(a|s)$ is achieved by updating its parameters $\theta$ using the policy gradient [36, Thm. 2]

$$\nabla J(\pi_\theta) = \mathbb{E}_{\pi_\theta} \left[ \nabla_\theta \log \pi_\theta(a|s) A_{\pi_\theta}(a, s) \right], \tag{1}$$

where $A_{\pi_\theta}(a, s)$ is the advantage function, which represents the expected improvement in reward by taking action $a$ in state $s$ compared to the average action taken in that state according to the policy $\pi_\theta(a|s)$. An estimation of the advantage $A_{\pi_\theta}(a, s)$ is usually provided by a neural network.

## 3.2 System model and set representations

We consider general continuous-time systems of the form

$$\dot{s} = f(s, a, w), \tag{2}$$

where $w \in \mathcal{W} \subset \mathbb{R}^{n^{\mathcal{W}}}$ denotes a disturbance. The input is piece-wise constant with a sampling interval $\Delta t$. We assume $\mathcal{S}$, $\mathcal{A}$, and $\mathcal{W}$ to be convex.

Zonotopes are a convex set representation well-suited for representing the relevant action set, due to the efficiency of computing their Minkowski sums and linear maps. A zonotope $\mathcal{Z} \subset \mathbb{R}^N$ with center $c \in \mathbb{R}^N$, generator matrix $G \in \mathbb{R}^{N \times P}$, and scaling factors $\beta \in \mathbb{R}^P$ is defined as

$$\mathcal{Z} = \left\{ c + G\beta \mid \|\beta\|_\infty \leq 1 \right\} = \langle c, G \rangle_{\mathcal{Z}}. \tag{3}$$

Additionally, let us denote that $G_{(\cdot, i)}$ returns the $i$-th column vector of the generator matrix $G$. The Minkowski addition $\mathcal{Z}_1 \oplus \mathcal{Z}_2$ of two zonotopes $\mathcal{Z}_1$, $\mathcal{Z}_2$ and the linear map $M\mathcal{Z}_1$ of a zonotope $\mathcal{Z}_1$ are given by [2, Eq. 2.1]

$$\mathcal{Z}_1 \oplus \mathcal{Z}_2 = \langle c_1 + c_2, [G_1 \quad G_2] \rangle_{\mathcal{Z}}, \tag{4a}$$

$$M\mathcal{Z}_1 = \langle Mc_1, MG_1 \rangle_{\mathcal{Z}}. \tag{4b}$$

# 4 Continuous action masking

To apply action masking, a relevant action set $\mathcal{A}^r(s) \subseteq \mathcal{A}$ has to be available, which constrains the action space $\mathcal{A}$ based on task knowledge. Let us denote the state-dependent relevant action set as $\mathcal{A}^r(s) \subseteq \mathcal{A}$. From now on, we omit the dependency on the state to simplify notation. For our continuous action masking methods, we specifically require the following two assumptions:

**Assumption 1.** *The relevant action set $\mathcal{A}^r$ is convex and its center and boundary points are computable.*

**Assumption 2.** *The policy $\pi_\theta : S \times \mathcal{A} \to \mathbb{R}_+$ of the agent is represented by a parameterized probability distribution $a \sim \pi_\theta(a|s)$.*

Common convex set representations that fulfill Assumption 1 are polytopes or zonotopes. Our continuous action masking methods transform the policy $\pi_\theta(a|s)$, for which the parameters $\theta$ usually specify a neural network, into the relevant policy $\pi_\theta^r : S \times \mathcal{A}^r \to \mathbb{R}_+$ through a functional $h : (\Pi \times \mathcal{P}(\mathcal{A})) \to \Pi^r$. Here, $\Pi$ is the space of all policies, $\Pi^r$ is the space of all relevant policies, and $\mathcal{P}(\mathcal{A})$ is the power set of all $\mathcal{A}^r$. The transformation is defined as

$$a^r \sim \pi_\theta^r(a^r|s) = h\big(\pi_\theta(a|s), \mathcal{A}^r\big), \tag{5}$$

and thereby ensures that $a^r \in \mathcal{A}^r$ always holds. Please note that policy gradient methods only require the ability to sample from and to compute the gradient of the policy distribution. Therefore, explicit closed-form expressions for $\pi_\theta^r(a^r|s)$ and $h$ are not necessary.

In the following subsections, we introduce and evaluate three masking methods: generator mask, ray mask, and distributional mask, as shown in Fig. 1. We derive the effect of each masking approach on the gradient of the objective function for stochastic policy gradient methods.

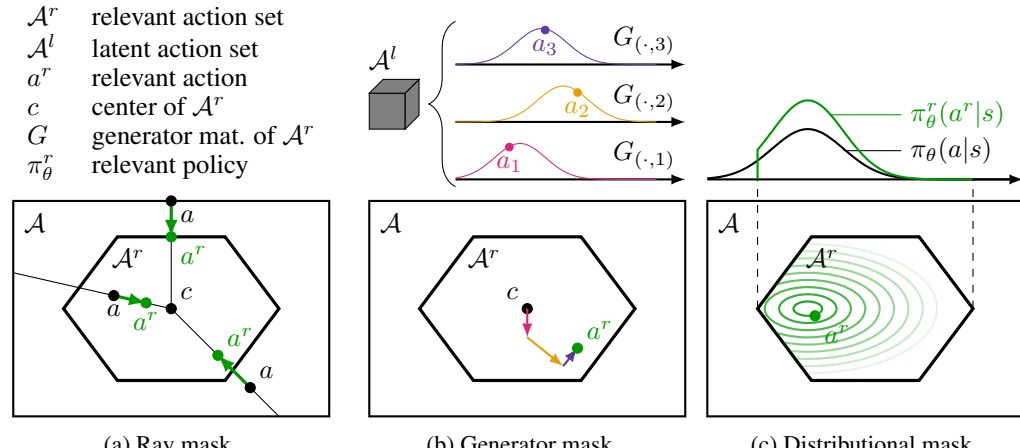

| | |
|---|---|
| $\mathcal{A}^r$ | relevant action set |
| $\mathcal{A}^l$ | latent action set |
| $a^r$ | relevant action |
| $c$ | center of $\mathcal{A}^r$ |
| $G$ | generator mat. of $\mathcal{A}^r$ |
| $\pi_\theta^r$ | relevant policy |

(a) Ray mask      (b) Generator mask      (c) Distributional mask

Figure 1: Illustration of masking methods in action space $\mathcal{A}$ with a hexagon-shaped relevant action set $\mathcal{A}^r$. The ray mask radially maps the actions towards the center of the relevant action set. The generator mask employs the latent action space $\mathcal{A}^l$, which is the generator space of the zonotope modeling the relevant action set. The distributional mask augments the policy probability density function so that it is zero outside the relevant action set.

## 4.1 Ray mask

The ray mask maps the action set $\mathcal{A}$ to $\mathcal{A}^r$ by scaling each action $a$ alongside a ray from the center of $\mathcal{A}^r$ to the boundary of $\mathcal{A}$, as shown in Figure 1a. Specifically, the relevant policy $\pi_\theta^r$ results from mapping the action $a$, sampled from $\pi_\theta(a|s)$, using the function $g_\mathrm{R} : \mathcal{A} \to \mathcal{A}^r$:

$$a^r = g_\mathrm{R}(a) = c + \frac{\lambda_{\mathcal{A}^r}(a)}{\lambda_{\mathcal{A}}(a)}(a - c). \tag{6}$$

Here, $\lambda_{\mathcal{A}}(a)$ and $\lambda_{\mathcal{A}^r}(a)$ denote the distances of the action $a$ to the boundaries of the relevant action set and action space, respectively, measured from the center of the relevant action set $c$ in the direction of $a$. Note that computing $\lambda_{\mathcal{A}^r}(a)$ for a zonotope requires solving a convex optimization problem, as specified in Appendix A.4. Yet, the ray mask is applicable for all convex sets, for which we can compute the center and boundary points. Since $g_\mathrm{R}(a)$ is bijective (see Appendix A.1 for a detailed proof), we can apply a change of variables [5, Eq. 1.27] to compute the relevant policy

$$\pi_\theta^r(a^r|s) = \pi_\theta\left(g_\mathrm{R}^{-1}(a^r)|s\right)\left|\det\left(\frac{\mathrm{d}}{\mathrm{d}a^r}g_\mathrm{R}^{-1}(a^r)\right)\right|, \tag{7}$$

where $g_\mathrm{R}^{-1}(a^r) = a$ is the inverse of $g_\mathrm{R}$. In general, there is no closed form of the distribution $\pi_\theta^r$ available. However, for stochastic policy gradient-based RL, we only require to sample from $\pi_\theta^r$ and compute its policy gradient. Samples from $\pi_\theta^r(a^r|s)$ are created by sampling from the original policy $a \sim \pi_\theta(a|s)$ followed by computing $a^r = g_\mathrm{R}(a)$. The policy gradient is derived next.

**Proposition 1.** *Policy gradient for the ray mask.*

The policy gradient of $\pi_\theta^r(a^r|s)$ for the ray mask is

$$\nabla_\theta J\left(\pi_\theta^r(a^r|s)\right) = \mathbb{E}_{\pi_\theta^r}\left[\nabla_\theta \log \pi_\theta(a|s) A_{\pi_\theta^r}(a^r, s)\right], \tag{8}$$

where $A_{\pi_\theta^r}(a^r, s)$ is the advantage function associated with $\pi_\theta^r(a^r|s)$.

*Proof.* The determinant in (7) is independent of $\theta$, i.e.,

$$\nabla_\theta \det\left(\frac{\mathrm{d}}{\mathrm{d}a^r}g_\mathrm{R}^{-1}(a^r)\right) = 0, \tag{9}$$

which simplifies the score function of $\pi_\theta^r(a^r|s)$ to

$$\nabla_\theta \log \pi_\theta^r(a^r|s) = \nabla_\theta \log \pi_\theta(a|s). \tag{10}$$

Combining (10) and the general form of the policy gradient in (1) for $\pi_\theta^r(a^r|s)$ results in (8).

$\square$

## 4.2 Generator mask

Zonotopes can be interpreted as the map of a hypercube in the generator space to a lower-dimensional space (see Section 3.2). The generator mask is based on exploiting this interpretation by letting the RL agent select actions in the hypercube of the generator space. Since the size of the output layer of the policy network cannot change during the learning process, we fix the dimension of the generator space. The generator mask requires the following assumption:

**Assumption 3.** *The relevant action set $\mathcal{A}^r(s)$ is represented by a zonotope $\langle c(s), G(s) \rangle_{\mathcal{Z}}$, with $G \in \mathbb{R}^{N \times P}$ and $c \in \mathbb{R}^N$, and a state-invariant number of generators $P$.*

Note that in practice, Assumption 3 can often be trivially fulfilled by choosing sufficiently many generators, and the number of generators $P$ is usually the output dimension of the parametrized policy. The domain of the policy $\pi_\theta(a|s)$ is the hypercube $\mathcal{A}^l = [-1,1]^P$, which can be interpreted as a latent action space, and the domain of the relevant policy $\pi_\theta^r(a^r|s)$ is a subset of the action space $\mathcal{A}^r \subseteq \mathcal{A}$ (see Figure 1b).

To derive the policy gradient of the generator mask, we assume:

**Assumption 4.** $\pi_\theta(a|s)$ *is a parametrized normal distribution* $\mathcal{N}(a; \mu_\theta, \Sigma_\theta)$.

**Proposition 2.** *The relevant policy of the generator mask is*

$$\pi_\theta^r(a|s) = \mathcal{N}(a; G\mu_\theta + c, G\Sigma_\theta G^T). \tag{11}$$

*Proof.* The generator mask $g_G : \mathcal{A}^l \to \mathcal{A}^r$ is:

$$a^r = g_G(a) = c + Ga, \tag{12}$$

which is a linear function. Therefore, the proof directly follows from the linear transformation of multivariate normal distributions [Thm. 3.3.3][40]. $\square$

Note that the ray mask and generator mask are mathematically equivalent to the approach in [19] if the relevant action set is constrained to intervals. Since $g_G(a)$ is not bijective in general, we cannot derive the gradient through a change of variables, as for the ray mask.

**Proposition 3.** *The policy gradient for $\pi_\theta^r(a^r|s)$ as defined in (11) with respect to $\mu_\theta$ and $\Sigma_\theta$ is*

$$\nabla_{\mu_\theta} \log \pi_\theta^r(a^r|s) = G^T(G\Sigma_\theta G^T)^{-1}(a^r - c - G\mu_\theta), \tag{13}$$

$$\nabla_{\Sigma_\theta} \log \pi_\theta^r(a^r|s) = -\frac{1}{2}\big(G^T(G\Sigma_\theta G^T)^{-1}G - G^T(G\Sigma_\theta G^T)^{-1}(a^r - c - G\mu_\theta) \tag{14}$$
$$(a^r - c - G\mu_\theta)^T(G\Sigma_\theta G^T)^{-1}G\big).$$

*Proof.* The proposition is proven in Appendix A.2. $\square$

Note that for the special case where $G^{-1}$ exists, i.e., $G$ is square and non-singular, the expressions in (13) and (14) simplify to $\nabla_{\mu_\theta} \log \pi_\theta(a|s)$, and $\nabla_{\Sigma_\theta} \log \pi_\theta(a|s)$, respectively, when Assumption 4 holds (see Proposition 5 in Appendix A.2). While the generator matrix will commonly not be square, as usually $P > N$, there are cases where $P = N$ is a valid choice, e.g., if there is a linear dependency between the action dimensions as for the 2D Quadrotor dynamics (see (40)).

## 4.3 Distributional mask

The intuition behind the distributional mask comes from discrete action masking, where the probability for irrelevant actions is set to 0 [15]. For continuous action spaces, we aim to achieve the same by ensuring that actions are only sampled from the relevant action set $\mathcal{A}^r$, by setting the density values of the relevant policy distribution $\pi_\theta^r(a^r|s)$ to zero for actions outside the relevant action set (see Figure 1c). For the one-dimensional case, this can be expressed by the truncated distribution [7]. In higher dimensions, the resulting policy distribution is

$$\pi_\theta^r(a^r|s) = \frac{\phi(a,s)\pi_\theta(a|s)}{\int_{\mathcal{A}^r} \pi_\theta(\tilde{a}|s)\mathrm{d}\tilde{a}}, \tag{15}$$

where $\phi(a, s)$ is the indicator function

$$\phi(a, s) = \begin{cases} 1 & \text{if } a \in \mathcal{A}^r, \\ 0 & \text{otherwise.} \end{cases} \tag{16}$$

Since there is no closed form of this distribution, we employ Markov chain Monte Carlo sampling to sample actions from the policy. More specifically, we utilize the random direction hit-and-run algorithm: a geometric random walk that allows sampling from a non-negative, integrable function $f : \mathbb{R}^N \to \mathbb{R}_+$, while constraining the samples to a bounded set [44]. The algorithm iteratively chooses a random direction from the current point, computes the one-dimensional, truncated probability density of $f$ along this direction, and samples a new point from this density. The approach is particularly effective for high-dimensional spaces where other sampling methods might struggle with convergence or efficiency. For the distributional mask, $f$ is the policy $\pi_\theta(a|s)$, and $\mathcal{A}^r$ is the set. As suggested by [22], we execute $N^3$ iterations before accepting the sample. To estimate the integral in (15), we use numerical integration with cubature [11], which is a method to approximate the definite integral of a function $l : \mathbb{R}^N \to \mathbb{R}$ over a multidimensional geometric set.

**Proposition 4.** *The policy gradient for the distributional mask is*

$$\nabla_\theta \log \pi_\theta^r(a^r|s) = \nabla_\theta \log \pi_\theta(a|s) - \nabla_\theta \log \int_{\mathcal{A}^r} \pi_\theta(\tilde{a}|s) \mathrm{d}\tilde{a}. \tag{17}$$

*Proof.* Equation (17) is obtained by calculating the gradient of the logarithm of (15). The indicator function $\phi(a, s)$ is not continuous and differentiable, which would necessitate the use of the sub-gradient for learning. However, since $a^r \in \mathcal{A}^r$ always holds, the gradient has to be computed for the continuous part of $\pi_\theta^r(a^r|s)$ only, and thus $\phi(a, s)$ can be omitted. $\qquad\square$

Since the gradient of the numeric integral $\int_{\mathcal{A}^r} \pi_\theta(\tilde{a}|s) \mathrm{d}\tilde{a}$ is intractable for zonotopes, we treat the integral as a constant in practice, and use $\nabla_\theta \log \pi_\theta^r(a^r|s) \approx \nabla_\theta \log \pi_\theta(a|s)$. We discuss potential improvements in Section 5.3.

## 5 Numerical experiments

We compare the three continuous action masking methods in four different environments: The simple and intuitive Seeker Reach-Avoid environment, the 2D Quadrotor environment to demonstrate the generalization from the continuous action masking approach in [19], and the 3D Quadrotor and Mujoco Walker2D environment to show the applicability to action spaces of higher dimension. Because the derivation of the relevant action set is not trivial in practice, we selected four environments for which we could compute intuitive relevant action sets. In particular, for the Seeker and Quadrotor environments, the relevant action set is a safe action set since it is computed so that the agent does not collide (Seeker) or leave a control invariant set (2D and 3D Quadrotor). For the Walker2D environment, the relevant action set is state-independent and models a power supply constraint for the actuators.

For the experiments, we extend the stable-baseline3 [29] implementation of proximal policy optimization (PPO) [33] by our masking methods. PPO is selected because it is a widely used algorithm in the field of RL and fulfills both Assumptions 2 and 4 by default. Apart from the masking agents, we also train a baseline agent with standard PPO that uses the action space $\mathcal{A}$ and a replacement agent, for which an action outside of $\mathcal{A}^r$ is replaced by a uniformly sampled action from $\mathcal{A}^r$ (see [19] for details). The replacement agent is an appropriate comparison to the masking agents since only relevant actions are executed while the replacement is implemented as part of the environment, which is usually easier than an implementation as part of the policy as for the masking methods. We conduct a hyperparameter optimization with 50 trials for each masking method and environment. The resulting hyperparameters are reported in Appendix A.9. All experiments are run on a machine with a Intel(R) Xeon(R) Platinum 8380 2.30 GHz processor and 2 TB RAM.

### 5.1 Environments

We briefly introduce the three environments and their corresponding relevant action sets $\mathcal{A}^r$. Parameters and dynamics for the environments are detailed in Appendix A.5.

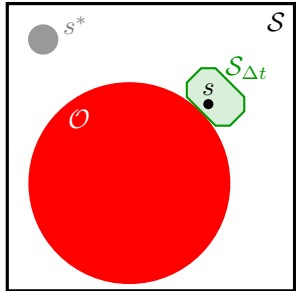 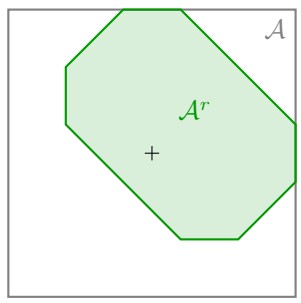

$\mathcal{A}$      action space
$\mathcal{A}^r$      relevant action set
$\mathcal{S}$      state space
$\mathcal{S}^r$      relevant state set
        $= \mathcal{S} \setminus \mathcal{O}$
$\mathcal{O}$      obstacle
$s$      agent position
$s^*$      goal position

Figure 2: The Seeker Reach-Avoid environment with state and action space. The agent (black) has to reach the goal (gray) while avoiding the obstacle (red). The center of the action space is illustrated by a cross and the relevant action set $\mathcal{A}^r$ for the current state is shown in green. The state set reachable at the next time step, by the relevant action set, is $\mathcal{S}_{\Delta t}$.

### 5.1.1 Seeker Reach-Avoid

This episodic environment features an agent navigating a 2D space, tasked with reaching a goal while avoiding a circular obstacle (see Figure 2). It is explicitly designed to provide an intuitive relevant action set $\mathcal{A}^r$. The system is defined as in Section 3.2: The dynamics for the position of the agent $s = [s_x, s_y]^T$ and the action $a = [a_x, a_y]^T$ is $\dot{s} = a$, and there are no disturbances.

The environment is characterized by the position of the agent, the goal position $s^*$, the obstacle position $o$, and the obstacle radius $r_o$. These values are pseudo-randomly sampled at the beginning of each episode, with the constraints that the goal cannot be inside the obstacle and the obstacle blocks the direct path between the initial position of the agent and the goal. The reward for each time step is

$$r(a, s) = \begin{cases} 100 & \text{if goal reached,} \\ -100 & \text{if collision occurred,} \\ -1 - \|s^* - s\|_2 & \text{otherwise.} \end{cases} \tag{18}$$

We compute the relevant action set $\mathcal{A}^r$ so that all actions that cause a collision with the obstacle or the boundary are excluded (see Appendix A.3.3).

### 5.1.2 2D Quadrotor

The 2D Quadrotor environment models a stabilization task and employs an action space where the two action dimensions are coupled, i.e., rotational movement is originating from differences between the action values and vertical movement is proportional to the sum of the action values. The relevant action set is computed based on the system dynamics and a relevant state set (see Appendix, Eq. (33)). The reward function is defined as

$$r(a, s) = \exp\left(-\|s - s^*\|_2 - \frac{0.01}{2} \left\| \left[\frac{a_1 - a_{1,\min}}{a_{1,\text{range}}}, \frac{a_2 - a_{2,\min}}{a_{2,\text{range}}}\right] \right\|_1 \right), \tag{19}$$

where $s^* = \mathbf{0}$ is the stabilization goal state, $a = [a_1, a_2]$ is the two-dimensional action, $a_{i,\min}$ is the lower bound for the actions in dimension $i$, and $a_{i,\text{range}}$ is the absolute difference between the lower and upper bound for the actions in dimension $i$.

### 5.1.3 3D Quadrotor

The third environment models a stabilization task for a quadrotor defined in [16]. The quadrotor has four action dimensions. We use the same reward (see (19)) and the same calculation approach for the relevant action set (see Appendix, Eq. (33)) as for the 2D Quadrotor.

### 5.1.4 Mujoco Walker2D

The relevant action sets of the two Quadrotor and the Seeker environments are sets that only contain safe actions for the current state. Computing safe action sets requires considerable domain knowledge

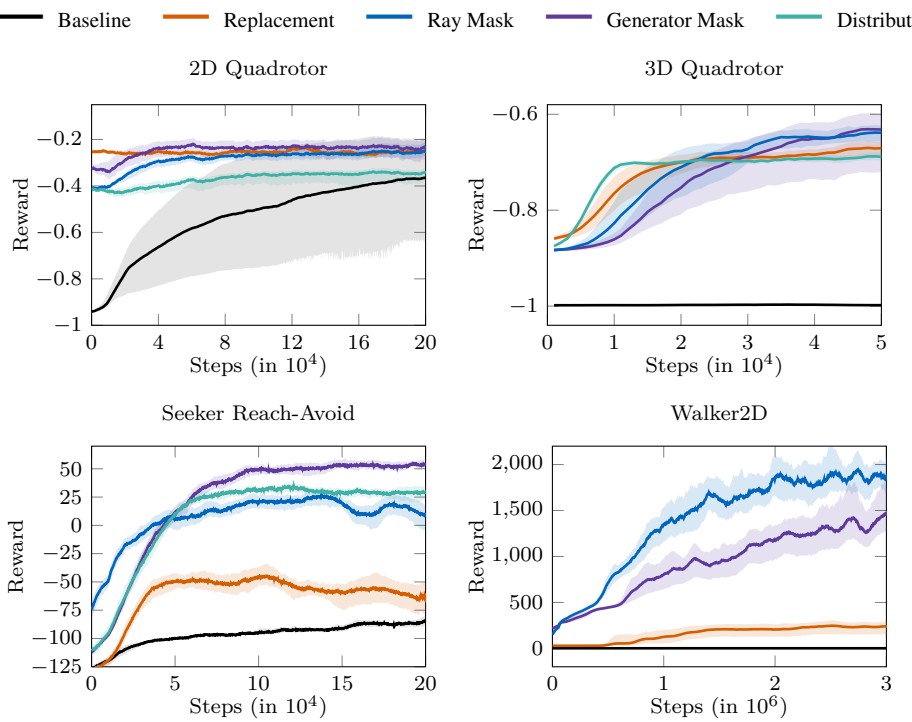

Figure 3: Average reward curves for benchmarks with transparent bootstrapped 95% confidence interval.

and for our case solving an optimization problems (see Appendix A.3). However, action masking is not restricted to safe relevant action sets, which we aim to demonstrate on the Mujoco Walker2D environment [39]. We extend the environment with a termination criterion, which ends an episode, when the the action violates the constraint $\|a\|_2 \leq \alpha_p$. This can be viewed as constraining the cumulative power output on all joints to a maximum value $\alpha_p$. We motivate this constraint by having a battery with a maximum power output that should not be exceeded in practice. Accordingly, we define the relevant action set as the static set

$$\mathcal{A}^r = \big\{a\big|\|a\|_2 \leq \alpha_p\big\}. \tag{20}$$

We under-approximate this relevant action set with a zonotope that consists of 36 generators.

## 5.2 Results

The reported training results are based on ten random seeds for each configuration. Figure 3 shows the mean reward and the bootstrapped 95% confidence intervals [27] for the four environments and Table 1 depicts the mean and standard deviation of the episode return during deployment. First, we present the results for the Seeker and Quadrotor environments for which the relevant action set is state-dependent and only includes safe actions. Then, we detail the results for the Walker2D environment with a static relevant action set.

For the Seeker and Quadotor environments, the baseline, i.e., PPO, converges significantly slower or not at all (see Figure 3). Additionally, the initial rewards when using action masking are significantly higher than for the baseline, indicating that the exploration is indeed constrained to relevant task-fulfilling actions. During deployment (see Table 1), the baseline episode returns are significantly lower than for action masking, while the three masking methods perform similarly. More specifically, for the Seeker environment, the relative volume of the relevant action set compared to the global action space is on average 70%, and all action masking methods converge significantly faster to high rewards than the baseline. The generator mask and the distributional mask achieve the highest final reward. Further, action replacement performs better than the baseline but significantly worse than

Table 1: Mean and standard deviation of episode return for ten runs per trained model.

| | Seeker | 2D Quad. | 3D Quad. | Walker2D |
|---|---|---|---|---|
| Baseline | $-71.03 \pm 19.67$ | $-0.80 \pm 0.15$ | $-1.00 \pm 0.00$ | $1.19 \pm 0.06$ |
| Replacement | $-60.21 \pm 19.98$ | $-0.25 \pm 0.03$ | $-0.68 \pm 0.09$ | $234.93 \pm 129.5$ |
| Ray | $-20.45 \pm 16.39$ | $-0.26 \pm 0.05$ | $-0.63 \pm 0.03$ | $1941.82 \pm 992.83$ |
| Generator | $18.60 \pm 22.02$ | $-0.25 \pm 0.02$ | $-0.68 \pm 0.07$ | $1443.62 \pm 702.7$ |
| Distributional | $-13.66 \pm 19.97$ | $-0.23 \pm 0.02$ | $-0.66 \pm 0.02$ | $-$ |

masking. For the 2D Quadrotor, the relative volume is on average $28\%$, which is much smaller than for the Seeker environment. In the 2D Quadrotor environment, the ray mask, generator mask, and action replacement achieve the highest reward. While the baseline converges significantly slower, the final average reward is similar to the one of the distributional mask. Please note that the confidence interval for the baseline is significantly larger, because three of the ten runs do not converge and, on average, exhibit a reward of one throughout training. Additionally, we observed that if we constrain the relevant action set to intervals for this environment, our optimization problem in (33) often renders infeasible, because the maximal relevant action set cannot be well approximated by an interval. Thus, the masking method proposed in [19] is not suitable for the 2D Quadrotor task. The results on the 3D Quadrotor are similar to those on the 2D Quadrotor; again, the generator mask converges the fastest but yields a final reward similar to that of the ray mask and action replacement. The relative volume of the relevant action set to the global action space is on average $25\%$. Notably, in this environment, the baseline does not learn a meaningful policy. Based on these three environments with state-dependent relevant action sets, it seems that action masking performs better than action replacement when the relative volume is not too small.

The training results of the Walker2D experiment are shown in Figure 3. While the generator and ray mask both learn a performant policy, the ray mask outperforms by a significant margin. The lower performance of the generator mask is likely due to the high-dimensional generator space with 36 dimensions. This is supported by initial experiments with 12 generators (i.e., a more conservative under-approximation of the $L_2$-norm) where the generator mask performed better compared to the ray mask. Replacement performs significantly worse than the two masking approaches, and the PPO baseline does not learn a meaningful policy since the environment is frequently reset due to violations of the power constraint in (20). The frequent terminations occur since in six action dimensions the relative volume of the unit ball compared to the unit box is $\approx 8\%$, i.e., more than $92\%$ of actions are violating the power constraint. To compare masking to a learning baseline, we also evaluated standard PPO without constraints, which performs slightly better than ray masking but almost always uses actions outside $\mathcal{A}^r$ (see Appendix A.7). The deployment results in Table 1 reflect similar results as the training; the ray mask achieves better rewards than the generator mask, followed by replacement, and the baseline performs the worst. We excluded the distributional mask for the Walker2D, since its computation time is approximately 170 times slower than the baseline, compared to a 1.6 increase for the generator, 2.7 for the ray mask, and 2.5 for action replacement (see Table 3). The severely increased computational cost for the distributional mask arises from the geometric random walks, which scale cubically with action dimensions.

### 5.3 Discussion and limitations

Our experimental results indicate that continuous action masking with zonotopes can improve both the sample efficiency and the final policy of PPO. While the sample efficiency is higher in our experiments, computing the relevant action set adds computational load as shown by the increased computation times (see Appendix A.6). Thus, in practice, a tight relevant action set might require more computation time than the additional samples for standard RL algorithms. Yet, if the relevant action set provides guarantees, e.g., is a set of verified safe actions, this increased computation time is often acceptable. Additionally, the computational effort for the masks differs. Given a relevant action zonotope, the generator mask adds a matrix-vector multiplication, which scales quadratically with action dimensions, the ray mask is dominated by the computation of the boundary points, which scales polynomially with action dimensions [20], and the distributional mask scales cubically with the dimension of the action space due to the mixing time of the hit-and-run algorithm [22]. Note that

for the Walker2D environment with six action dimensions, the computation time for the hit-and-run algorithm is so high that the distributional mask evaluation was infeasible.

The ray mask and generator mask are based on functions $g_R$ and $g_G$ that map to relevant actions. There are two different approaches of incorporating these functions into RL algorithms. One is to apply the mapping as part of the environment on an action that is sampled by a standard RL policy. The second option, which we use, is to consider the masking as part of the policy, which creates the relevant policy as in (5). This has three main benefits over integrating masking as part of the environment. First, the actions passed to the environment are better interpretable, e.g., for the generator mask, adding the masking mapping function to the environment leads to a policy that samples actions from the generator space, which commonly does not have an intuitive interpretation. Second, more formulations of the functional $h$ are possible, e.g., the distributional mask, and, third, the mapping function can be included in the gradient calculation. For mathematically sound masking as part of the policy, the correct gradient needs to be derived for each RL algorithm, and the standard RL algorithms need to be adapted accordingly. However, the empirical benefit could be minor. Thus, future work should investigate the significance of the correct gradient on a variety of tasks. Note that we showed in Proposition 1 that for the ray mask, the PPO gradient with respect to the original policy and relevant policy is the same. Thus, for the ray mask, it does not matter if it is viewed as part of the policy or the environment.

PPO is a common RL algorithm. However, off-policy algorithms such as Twin Delayed DDPG (TD3) [9] and Soft Actor-Critic (SAC) [12] are frequently employed as well. The ray mask and generator mask are conceptually applicable for deterministic policies as used in TD3. Yet, the implications on the gradient must be derived for each RL algorithm and are subject to future work. For the distributional mask, treating the integral in (15) as constant with respect to $\theta$ is a substantial simplification, which might be an explanation for the slightly worse convergence of the distributional mask since this introduces an off-policy bias. To address this in future work, one could approximate the integral with a neural network, which has the advantage that it is easily differentiable.

We focus this work on the integration of a convex relevant action set into RL and assume that an appropriate relevant action set can be obtained. While convex sets are a significant generalization from previous work [19], they might not be sufficient for some applications, e.g., tasks where relevant action sets are disjoint. Thus, future work could include investigating hybrid RL approaches [25] to increase the applicability to multiple convex sets or non-convex sets, such as constrained polynomial zonotopes [17]. Further, obtaining the relevant action set can be a major challenge in practice, in particular when the relevant action set is different for each state. Such a high state-dependency is likely when the notion of relevance is safety, while for other definitions of action relevance, the relevant action set might be easy to pre-compute, e.g., excluding high steering angles at high velocities. Additionally, there might be an optimal precision of the relevant action set due to two opposing mechanisms. On the one hand, the larger the relevant action set is with respect to the action space, the smaller the sample efficiency gain from action masking might get. On the other hand, a tight relevant action set might require significant computation time to obtain. Thus, future work should investigate efficient methods to obtain sufficiently tight relevant action sets.

## 6 Conclusion

We propose action masking methods for continuous action spaces that focus the exploration on the relevant part of the action set. In particular, we extend previous work on continuous action masking from using intervals as relevant action sets to using convex sets. To this end, we have introduced three masking methods and have derived their implications on the gradient of PPO. We empirically evaluated our methods on four benchmarks and observed that the generator mask and ray mask perform best. If the relevant action set can be described by a zonotope with fixed generator dimensions and the policy follows a normal distribution, the generator mask is straightforward to implement. If the assumptions for the generator mask cannot be fulfilled, the ray mask is recommended based on our experiments. Because of subpar performance and longer computation time, the distributional mask needs to be further improved. Future work should also investigate a broad range of benchmarks to identify the applicability and limits of continuous action masking with convex sets more clearly.

## Acknowledgments and Disclosure of Funding

We thank Matthias Killer for conducting preliminary experiments. We gratefully acknowledge that this project was funded by the Deutsche Forschungsgemeinschaft (DFG, German Research Foundation) – SFB 1608 – 501798263, AL 1185/9-1, AL 1185/33-1, and the Bavarian Research Foundation project STROM (Energy - Sector coupling and microgrids).

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

# A Appendix

## A.1 Proof of bijectivity of the ray mapping function.

**Lemma 1.** *The mapping function* $g : \mathcal{A} \to \mathcal{A}^r$ *of the ray mask* $g(a) = c + \frac{\lambda_{\mathcal{A}^r}(a)}{\lambda_{\mathcal{A}}(a)}(a - c)$ *is bijective.*

*Proof.* We show that $g(a)$ is bijective by showing that the function is both injective and surjective.

For any convex set $\mathcal{A}^r$ with center $c$, we can construct a ray from $c$ through any $a^r \in \mathcal{A}^r$ as $r_\gamma(t) = c + \gamma t$, where $\gamma = \frac{a-c}{\|a-c\|_2}$, and $t \in [0, t_{max}]$. Since $\mathcal{A}^r \subseteq \mathcal{A}$, this also holds $\forall a \in \mathcal{A}$. By construction, any two distinct rays only intersect in $c$. For all points on a given ray, the scaling factors $\lambda_{\mathcal{A}}$ and $\lambda_{\mathcal{A}^r}$ are constants, allowing us to rewrite $g(a)$ for all $a = r_\gamma(t)$ as the linear function:

$$g\left(r_\gamma(t)\right) = c + \frac{\lambda_{\mathcal{A}^r}}{\lambda_{\mathcal{A}}}\left(r_\gamma(t) - c\right).$$

To show that $g(a)$ is injective, i.e. $\forall a_1, a_2 \in \mathcal{A}$, if $a_1 \neq a_2 \implies g(a_1) \neq g(a_2)$, consider the following two cases for $a_1 \neq a_2$.

Case 1: If $a_1$ and $a_2$ are on the same ray, then, $a_1 = r_\gamma(t_1)$ and $a_2 = r_\gamma(t_2)$ with $t_1 \neq t_2$. Since $g\left(r_\gamma(t)\right)$ is linear and thereby monotonic, it follows that $g\left(r_\gamma(t_1)\right) \neq g\left(r_\gamma(t_2)\right)$ and consequently $g(a_1) \neq g(a_2)$.

Case 2: Otherwise, $a_1$ and $a_2$ are on different rays and $a_1 \neq a_2 \neq c$, so $g(a_1) \neq g(a_2)$ follows directly as the rays only intersect in $c$.

Thus, $g(a)$ is injective.

For showing $g(a)$ to be surjective, it is sufficient to show that $\forall a^r \in \mathcal{A}^r$, there exists an $a \in \mathcal{A}$, for which $g(a) = a^r$.

Consider the ray $r_\gamma(t)$ passing through $a^r$. We know that $g\left(r_\gamma(0)\right) = g(c) = c$, and $g\left(r_\gamma(t_{max})\right)$ is the boundary point of $\mathcal{A}^r$ from $c$ in the direction $d$. Moreover, $a^r$ lies on the line segment between $c$ and $g\left(r_\gamma(t_{\max})\right)$. Since $g\left(r_\gamma(t)\right)$ is linear and continuous, according to the intermediate value theorem, $\exists t^* \in [0, t_{\max}]$, for which $g\left(r_\gamma(t^*)\right) = a^r$ for any $a^r \in \mathcal{A}^r$ along the ray $r_\gamma(t)$. As we can construct such a ray through any point in $\mathcal{A}^r$, we have shown that for every $a^r \in \mathcal{A}^r$, there exists an $a \in \mathcal{A}$ such that $g(a) = a^r$, thus proving surjectivity. $\square$

## A.2 Policy gradient for the generator mask

The log probability density function of the relevant policy for the generator mask is

$$\log \pi_\theta^r(a^r|s) = -\frac{d}{2}\log(2\pi) - \frac{1}{2}\log|G\Sigma_\theta G^T| - \frac{1}{2}(a^r - c - G\mu_\theta)^T(G\Sigma_\theta G^T)^{-1}(a^r - c - G\mu_\theta). \quad (21)$$

We can derive the gradient w.r.t $\mu_\theta$ as

$$
\begin{aligned}
\nabla_{\mu_\theta} \log \pi_\theta^r(a^r|s) &= -\frac{1}{2}\nabla_{\mu_\theta}(a^r - c - G\mu_\theta)^T(G\Sigma_\theta G^T)^{-1}(a^r - c - G\mu_\theta) \\
&= -\frac{1}{2}\left(-2(G\Sigma_\theta G^T)^{-1}(a^r - c - G\mu_\theta)\right)\nabla_{\mu_\theta}(a^r - c - G\mu_\theta) \\
&= G^T(G\Sigma_\theta G^T)^{-1}(a^r - c - G\mu_\theta),
\end{aligned} \quad (22)
$$

and w.r.t $\Sigma_\theta$ as

$$
\begin{aligned}
\nabla_{\Sigma_\theta} \log \pi_\theta^r(a^r|s) &= \\
&= -\frac{1}{2}\nabla_{\Sigma_\theta}\log|G\Sigma_\theta G^T| - \frac{1}{2}\nabla_{\Sigma_\theta}(a^r - c - G\mu_\theta)^T(G\Sigma_\theta G^T)^{-1}(a^r - c - G\mu_\theta) \\
&= -\frac{1}{2}(G\Sigma_\theta G^T)^{-1}\nabla_{\Sigma_\theta}G\Sigma_\theta G^T \\
&\quad + \frac{1}{2}(G\Sigma_\theta G^T)^{-1}(a^r - c - G\mu_\theta)(a^r - c - G\mu_\theta)^T(G\Sigma_\theta G^T)^{-1}\nabla_{\Sigma_\theta}G\Sigma_\theta G^T \\
&= -\frac{1}{2}\left(G^T(G\Sigma_\theta G^T)^{-1}G - G^T(G\Sigma_\theta G^T)^{-1}(a^r - c - G\mu_\theta)(a^r - c - G\mu)^T(G\Sigma_\theta G^T)^{-1}G\right)
\end{aligned} \quad (23)
$$

We can state the following for the special case when the inverse of $G$ exists.

**Proposition 5.** *If $G$ is invertible, $\nabla_\theta \log \pi_\theta^r(a^r|s) = \nabla_\theta \log \pi_\theta(a|s)$ holds for the generator mask.*

*Proof.* If $G$ is invertible, we can further simplify (22) to

$$
\begin{aligned}
\nabla_{\mu_\theta} \log \pi_\theta^r(a^r|s) &= G^T G^{-T} \Sigma_\theta^{-1} G^{-1}(a^r - c - G\mu_\theta) \\
&= \Sigma_\theta^{-1} \left( G^{-1}(a^r - c) - G^{-1} G \mu_\theta \right) \\
&= \Sigma_\theta^{-1} (a - \mu_\theta) = \nabla_{\mu_\theta} \log \pi_\theta(a|s),
\end{aligned}
\tag{24}
$$

and (23) to

$$
\begin{aligned}
\nabla_{\Sigma_\theta} \log \pi_\theta^r(a^r|s) &= -\frac{1}{2} \left( \Sigma_\theta^{-1} - \Sigma_\theta^{-1} G^{-1}(a^r - c - G\mu_\theta)(a^r - c - G\mu_\theta)^T G^{-T} \Sigma_\theta^{-1} \right) \\
&= -\frac{1}{2} \left( \Sigma_\theta^{-1} - \Sigma_\theta^{-1}(a - \mu_\theta)(a - \mu_\theta)^T \Sigma_\theta^{-1} \right) = \nabla_{\Sigma_\theta} \log \pi_\theta(a|s),
\end{aligned}
\tag{25}
$$

by using $a = g^{-1}(a^r) = G^{-1}(a^r - c)$, and thus proving the statement. $\qquad \square$

### A.3 Computation of the relevant action set

#### A.3.1 General case

Given an initial state $s_0$, an input trajectory $a_{(\cdot)}$, and a disturbance trajectory $w_{(\cdot)}$, we denote the solution of (2) at time $t$ as $\xi_t(s_0, a_{(\cdot)}, w_{(\cdot)})$. Assuming a sampled controller with piecewise-constant input $a$, we define the reachable set of (2) after one time step $\Delta t$ given a set of initial states $\mathcal{S}_0$ and a set of possible inputs $\tilde{\mathcal{A}} \subseteq \mathcal{A}$ as

$$
\mathcal{R}_{\Delta t}^e(\mathcal{S}_0, \tilde{\mathcal{A}}) = \left\{ \xi_{\Delta t}(s_0, a, w_{(\cdot)}) \, \middle| \, \exists x_0 \in \mathcal{S}_0, \exists a \in \tilde{\mathcal{A}}, \forall t \colon \exists w_t \in \mathcal{W} \right\}.
\tag{26}
$$

The reachable set over the time interval $[0, \Delta t]$ is defined as

$$
\mathcal{R}_{[0,\Delta t]}^e(\mathcal{S}_0, \tilde{\mathcal{A}}) = \bigcup_{\tau \in [0,\Delta t]} \mathcal{R}_\tau^e(\mathcal{S}_0, \tilde{\mathcal{A}}).
\tag{27}
$$

For many system classes it is impossible to compute the reachable set exactly [28], which is why we generally compute overapproximations $\mathcal{R}(\cdot) \supseteq \mathcal{R}^e(\cdot)$.

To guarantee constraint satisfaction over an infinite time horizon, we use a relevant state set $\mathcal{S}^r$. In this work, we choose $\mathcal{S}^r$ to be a robust control invariant set in the sense that we can always find an input which guarantees that the reachable set at the next time step is contained in $\mathcal{S}^r$ and that all state constraints are satisfied in the time interval in between [32]:

$$
\exists a \in \mathcal{A} \colon \mathcal{R}_{\Delta t}(\mathcal{S}^r, a) \subseteq \mathcal{S}^r, \mathcal{R}_{[0,\Delta t]}(\mathcal{S}^r, a) \subseteq \mathcal{S}.
\tag{28}
$$

We compute the relevant action set as the largest set of inputs that allows us to keep the system in the relevant state set in the next time step. We define a parameterized action set $\mathcal{A}^r(p)$, where $p \in \mathbb{R}^{n^p}$ is a parameter vector. The relevant action set is then computed with the optimal program

$$
\begin{aligned}
\max_p \quad & \widetilde{\mathrm{Vol}}\left(\mathcal{A}^r(p)\right) \\
\text{subject to} \quad & \mathcal{A}^r(p) \subseteq \mathcal{A} \\
& \mathcal{R}_{\Delta t}(\mathcal{S}_0, \mathcal{A}^r(p)) \subseteq \mathcal{S}^r \\
& \mathcal{R}_{[0,\Delta t]}(\mathcal{S}_0, \mathcal{A}^r(p)) \subseteq \mathcal{S},
\end{aligned}
\tag{29}
$$

where $\widetilde{\mathrm{Vol}}(\cdot)$ is a proxy function for the volume in the sense that a maximization of $\widetilde{\mathrm{Vol}}(\cdot)$ is suboptimally maximizing the volume. In the following, we provide a detailed formulation of (29) as an exponential cone program.

### A.3.2 Exponential cone program

We consider the discrete-time linearization of our system

$$s_{k+1} = As_k + Ba_k + w'_k, \tag{30}$$

where $w'_k \in \mathcal{W}'(s_k)$ additionally contains linearization errors and the enclosure of all possible trajectory curvatures between the two discrete time steps [2]. Furthermore, we assume $\mathcal{S}, \mathcal{A}, \mathcal{W}$, and $\mathcal{S}^r$ to be zonotopes.

With regard to the input set parameterization, we consider a template zonotope with a predefined template generator matrix $\tilde{G}$. We use the vector $\tilde{p} \in \mathbb{R}^P_{>0}$ of generator scaling factors to scale the generator matrix. The parameterized template zonotope is then given by

$$\mathcal{A}^r(p) = \langle c, \tilde{G}\, diag(\tilde{p}) \rangle_{\mathcal{Z}}, \tag{31}$$

where $p = \begin{bmatrix} c & \tilde{p} \end{bmatrix}^\top$. In the following, we denote the center and generator matrix of a specific zonotope $\mathcal{Z}$ by $c^{\mathcal{Z}}$ and $G^{\mathcal{Z}}$, respectively. A zonotope $\mathcal{Z}_1$ is contained in a zonotope $\mathcal{Z}_2$ if there exist $\Gamma \in \mathbb{R}^{P^{\mathcal{Z}_2} \times P^{\mathcal{Z}_1}}$, $\beta \in P^{\mathcal{Z}_2}$, such that [31, Corollary 4]

$$G^{\mathcal{Z}_1} = G^{\mathcal{Z}_2}\Gamma$$
$$c^{\mathcal{Z}_2} - c^{\mathcal{Z}_1} = G^{\mathcal{Z}_2}\beta \tag{32}$$
$$\|[\Gamma, \beta]\|_\infty \leq 1.$$

Based on the linearized system dynamics in (30), the definitions from (4), and using $\mathcal{R}$ as a shorthand for $\mathcal{R}_{\Delta t}(\cdot)$, we have $G^{\mathcal{R}} = \begin{bmatrix} AG^{\mathcal{S}_0} & BG^{\mathcal{A}^r} & G^{\mathcal{W}} \end{bmatrix}$ and $c^{\mathcal{R}} = \begin{bmatrix} Ac^{\mathcal{S}_0} & Bc^{\mathcal{A}^r} & c^{\mathcal{W}} \end{bmatrix}$. Since we already consider trajectory curvature in the disturbance, we only need to guarantee that the relevant input set is contained in the feasible input set and that the reachable set of the next time step is contained in the relevant state set. Using the containment condition from (32), we define the auxiliary variables $\Gamma^{\mathcal{R}}, \beta^{\mathcal{R}}, \Gamma^{\mathcal{A}^r}$, and $\beta^{\mathcal{A}^r}$ and solve

$$\begin{aligned}
\max_p \quad & \widetilde{\texttt{Vol}}\left(\mathcal{A}^r(p)\right) \\
\text{subject to} \quad & G^{\mathcal{R}} - G^{\mathcal{S}^r}\Gamma^{\mathcal{R}} = \mathbf{0} \\
& c^{\mathcal{S}^r} - c^{\mathcal{R}} - G^{\mathcal{S}^r}\beta^{\mathcal{R}} = \mathbf{0} \\
& \|\begin{bmatrix} \Gamma^{\mathcal{R}} & \beta^{\mathcal{R}} \end{bmatrix}\|_\infty \leq 1 \\
& G^{\mathcal{A}^r} - G^{\mathcal{A}}\Gamma^{\mathcal{A}^r} = \mathbf{0} \\
& c^{\mathcal{A}} - c^{\mathcal{A}^r} - G^{\mathcal{A}}\beta^{\mathcal{A}^r} = \mathbf{0} \\
& \|\begin{bmatrix} \Gamma^{\mathcal{A}^r} & \beta^{\mathcal{A}^r} \end{bmatrix}\|_\infty \leq 1,
\end{aligned} \tag{33}$$

where $\mathbf{0}$ are zero matrices of appropriate dimensions. By vectorizing $\Gamma^{\mathcal{R}}$ and $\Gamma^{\mathcal{A}^r}$ and resolving the infinity norm constraints [4, Sec. 1.3], we can obtain a formulation with purely linear constraints. Since computing the exact volume of a zonotope is computationally expensive, we approximate it by computing the geometric mean of the parameterization vector:

$$\widetilde{\texttt{Vol}}(\mathcal{A}^r(p)) = \left(\prod_i \tilde{p}_i\right)^{\frac{1}{P}}. \tag{34}$$

With the cost function from (34), the problem in (33) is an exponential cone program, which is convex and can be efficiently computed [6, Sec. 2.5].

### A.3.3 Seeker Reach-Avoid environment

Because of the dynamics of the Seeker Reach-Avoid environment, we can simplify the relevant action set computation to the following optimization problem:

$$\begin{aligned}
\max_p \quad & \widetilde{\texttt{Vol}}(\mathcal{A}^r(p)) \\
\text{subject to} \quad & \mathcal{A}^r(p) \subseteq \mathcal{A} \\
& \mathcal{S}^r_{\Delta t} = s \oplus \mathcal{A}^r(p) \\
& \mathcal{S}^r_{\Delta t} \subseteq [-10, 10]^2 \\
& \mathcal{S}^r_{\Delta t} \cap \mathcal{O} = \emptyset,
\end{aligned} \tag{35}$$

where $\mathcal{S}_{\Delta t}$ is the reachable set of the agent in the next time step, the set $\mathcal{O}$ represents the obstacle, and the box $[-10, 10]^2$ is the outer boundary of the state space. The constraints represent three intuitive geometric constraints. First, containment of the relevant action set in the action set (usually, the interval $[-1, 1]^2$, second, the containment of the relevant state set of the next time step in the environment boundaries, and the last constraint enforces that there is no collision with the obstacle possible. To enforce these constraints for zonotopes, we use the support function [3, Lemma 1]

$$\rho_{\mathcal{Z}}(l) = l^T c + \sum_{i=1}^{P} |l^T G_{(\cdot, i)}|, \tag{36}$$

where $l$ is the vector in the direction of interest. For the first constraint, we need to ensure that the support function for the basis vectors $e_1$, and $e_2$ (and there negative versions) are less than or equal to 1, i.e.,

$$\begin{bmatrix} \rho_{\mathcal{A}^r}(e_1) \\ \rho_{\mathcal{A}^r}(e_2) \end{bmatrix} = \begin{bmatrix} c_1^{\mathcal{A}^r} + \sum_{i=1}^{P} |G_{(1,i)}^{\mathcal{A}^r}| \\ c_2^{\mathcal{A}^r} + \sum_{i=1}^{P} |G_{(2,i)}^{\mathcal{A}^r}| \end{bmatrix} = c^{\mathcal{A}^r} + \sum_{i=1}^{P} |G_{(\cdot,i)}^{\mathcal{A}^r}| \leq \begin{bmatrix} 1 \\ 1 \end{bmatrix}. \tag{37}$$

From (36) it is apparent that using the negative basis vectors $-e_1$ and $-e_2$, just flips the sign of $c^{\mathcal{A}^r}$ in (37). The constraints for $\mathcal{S}_{\Delta t}^r \subseteq [-10, 10]^2$ can be constructed similarly. For simplicity, we express the element-wise inequality in (37) using the $L_\infty$-norm, and write the optimization problem as

$$\begin{aligned}
\max_{p} \quad & \widetilde{\texttt{Vol}}(\mathcal{A}^r(p)) \\
\text{subject to} \quad & \|c^{\mathcal{A}^r} + \sum_i |G_{(\cdot,i)}^{\mathcal{A}^r}|\|_\infty \leq 1 \\
& \| - c^{\mathcal{A}^r} + \sum_i |G_{(\cdot,i)}^{\mathcal{A}^r}|\|_\infty \leq 1 \\
& \|c^{\mathcal{A}^r} + s + \sum_i |G_{(\cdot,i)}^{\mathcal{A}^r}|\|_\infty \leq 10 \\
& \| - c^{\mathcal{A}^r} + s + \sum_i |G_{(\cdot,i)}^{\mathcal{A}^r}|\|_\infty \leq 10 \\
& n^T(c^{\mathcal{A}^r} + s) + \sum_i |n^T G_{(\cdot,i)}^{\mathcal{A}^r}| \leq b.
\end{aligned} \tag{38}$$

The last constraint represents an under-approximation of $\mathcal{S}_{\Delta t}^r \cap \mathcal{O} = \emptyset$ enforcing the containment of $\mathcal{A}^r$ in the halfspace $\{x | n^T x \leq b\}$ through the support function $\rho_{\mathcal{A}^r}(n)$. The directional vector $n = \frac{o-s}{\|o-s\|}$ where $o$ is the center of the obstacle, and the offset $b = n^T(o - n)r$ where $r$ is the radius of the obstacle. This ensures that the intersection between the reachable set $\mathcal{S}_{\Delta t}^r$ and the obstacle $\mathcal{O}$ will be empty since the halfspace is constructed tangential to the obstacle at the intersection point of the line from the agent to the center of the obstacle.

### A.4 Computation of zonotope boundary points

The boundary point $p \in \mathbb{R}^N$ on a zonotope $\langle c, G \rangle_{\mathcal{Z}} \subset \mathbb{R}^N$ in a direction $d \in \mathbb{R}^N$ starting from a point $x \in \mathbb{R}^N$ is obtained by solving the linear program

$$\begin{aligned}
\min_{\alpha \in \mathbb{R}, \gamma \in \mathbb{R}^N} \quad & \alpha \\
\text{subject to} \quad & x + \alpha d = c + G\gamma \\
& \|\gamma\|_\infty \leq 1
\end{aligned} \tag{39}$$

and computing $p = x + \alpha d$ [20].

### A.5 Parameters and dynamics for environments

We provide the action space bounds, the state space bounds, and the generator template matrix $\tilde{G}$ for the Seeker and Quadrotor environments in Table 2.

Table 2: Parameters for important sets of environments.

| Parameter | Seeker | 2D Quadrotor | 3D Quadrotor |
|---|---|---|---|
| Lower bound actions | [-1, -1] | [6.83, 6.83] | [-9.81, -0.5, -0.5, -0.5] |
| Upper bound actions | [1, 1] | [8.59, 8.59] | [2.38, 0.5, 0.5, 0.5] |
| Lower bound states | [-10, -10] | [-1.7, 0.3, -0.8, ... -1, $-\pi/12$, $-\pi/2$] | [-3, -3, -3, -3, -3, -3, ... $-\pi/4$, $-\pi/4$, $-\pi$, -3, -3, -3] |
| Upper bound states | [10, 10] | [1.7, 2.0, 0.8, ... 1.0, $\pi/12$, $\pi/2$] | [3, 3, 3, 3, 3, 3, ... $\pi/4$, $\pi/4$, $\pi$, 3, 3, 3] |
| Template matrix $\tilde{G}$ | $\begin{bmatrix} 1 & 1 & 1 & 0 \\ 1 & -1 & 0 & 1 \end{bmatrix}$ | $\begin{bmatrix} 1 & 1 & 1 \\ 1 & -1 & 0 \end{bmatrix}$ | $I_4$ |

The system dynamics of the 2D Quadrotor is:

$$
\dot{s} = \begin{pmatrix} \dot{x} \\ \dot{z} \\ (a_1 + a_2)k\sin(\theta) \\ -g + (a_1 + a_2)k\cos(\theta) \\ \dot{\theta} \\ -d_0\theta - d_1\dot{\theta} + n_0(-a_1 + a_2) \end{pmatrix} + \begin{pmatrix} 0 \\ 0 \\ w_1 \\ w_2 \\ 0 \\ 0 \end{pmatrix}, \tag{40}
$$

where the state is $s = \left[x, z, \dot{x}, \dot{z}, \theta, \dot{\theta}\right]^T$, the action is $a = [a_1, a_2]^T$, and the disturbance is $w = [w_1, w_2]^T$. The dynamics are adapted from [24, Eq. 43] so that the actions are two independent thrusts. Additionally, the parameters used in our experiments are $g = 9.81\,\mathrm{m\,s^{-2}}$, $k = 1\,1/\mathrm{kg}$, $d_0 = 70$, $d_1 = 17$, $n_0 = 55$, and $\mathcal{W} = [-0.08, -0.08] \times [0.08, 0.08]$.

The 3D Quadrotor is modeled in a twelve-dimensional state space with state $s = [x, y, z, \dot{x}, \dot{y}, \dot{z}, \theta, \phi, \psi, \dot{\theta}, \dot{\phi}, \dot{\psi}]^T$ and four-dimensional action space with action $a = [a_1, a_2, a_3, a_4]^T$. The system dynamics are $\dot{s} = [\dot{x}, \dot{y}, \dot{z}, -9.81\phi, 9.81\theta, a_1, \dot{\theta}, \dot{\phi}, \dot{\psi}, a_2, a_3, a_4]^T$ [16].

For the Walker2D environment, we use the standard parameters of the gymnasium implementation[2]. The relevant action set is a zonotope under-approximation of the relevant action set stated in (20) with 36 generators and $\alpha_P = 1$.

## A.6  Computational time for training

Note that the increased computation time of the masking approaches compared to the baseline is mainly due to the computation of the relevant action sets. Since the relevant action set is a zonotope, the generator mask does not require expensive additional computations. The increased runtime for the ray mask mainly originates from the computation of the boundary points (see Appendix A.4). For the distributional mask, the increased runtime is mostly caused by the Markov chain Monte Carlo sampling of the action.

Table 3: Mean and standard deviation of runtime in hours for training runs on the utilized machine.

| | Seeker | 2D Quadrotor | 3D Quadrotor | Walker2D |
|---|---|---|---|---|
| Baseline | $0.045 \pm 0.001$ | $1.049 \pm 0.197$ | $0.493 \pm 0.077$ | $0.350 \pm 0.007$ |
| Ray mask | $0.865 \pm 0.013$ | $3.140 \pm 0.097$ | $2.031 \pm 0.258$ | $0.962 \pm 0.043$ |
| Generator mask | $0.699 \pm 0.016$ | $2.533 \pm 0.021$ | $1.528 \pm 0.010$ | $0.557 \pm 0.021$ |
| Distributional mask | $1.620 \pm 0.017$ | $4.083 \pm 0.080$ | $3.130 \pm 0.045$ | $\approx 60$ |
| Replacement | $0.823 \pm 0.015$ | $2.782 \pm 0.048$ | $1.765 \pm 0.033$ | $0.880 \pm 0.039$ |

---

[2]https://gymnasium.farama.org/environments/mujoco/walker2d/

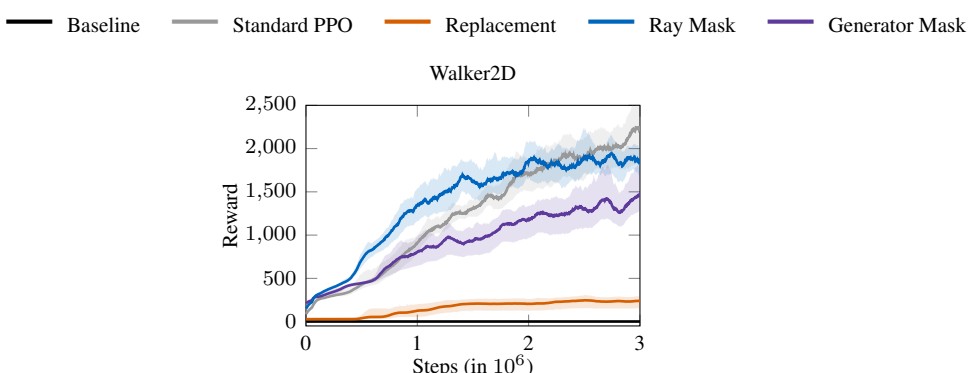

Figure 4: Average reward curves for Walker2D with transparent bootstrapped 95% confidence interval including standard PPO as additional baseline.

## A.7 Quantitative results for Walker2D environment compared to standard PPO

If we remove the power constraint that resets the environment whenever an action outside of $\mathcal{A}^r$ is selected, the standard PPO is able to learn a policy that performs slightly better than the ray mask towards the end of training while converging slightly slower (see Figure 4). However, during deployment, only $0.05\%$ of the actions from this policy are from within $\mathcal{A}^r$. Yet, the reward that standard PPO achieves during deployment is on average $2177.57$ with standard deviation $1238.42$. This is slightly higher than the deployment results for the ray mask (see Table 1).

## A.8 Qualitative results for the Seeker environment

Figure 5 presents ten example trajectories for a trained agent using each of the five approaches under consideration. Notably, the continuous action masking agents solve the task by safely reaching the goal while the PPO baseline still collides with the obstacle. The replacement approach is ineffective, with the agent frequently failing to reach the goal. In fact, for the ten displayed rollouts, no replacement agent reaches the goal. This discrepancy with respect to the training performance where the replacement agent reaches the goal frequently is due to using the policy in a deterministic setting for deployment. The simple dynamics of the Seeker environment make it possible to directly visualize the relevant action set $\mathcal{A}^r$ at each time step along the trajectory, which is depicted in the lower half of Figure 5. For the generator mask, the least amount of $\mathcal{A}^r$ is plotted, which indicates that the generator mask agent reaches the goal most efficiently, i.e., the lowest amount of steps are required.

## A.9 Hyperparameters for learning algorithms

We specify the hyperparameters for the three masking approaches, replacement and baseline in the Seeker (Table 4), the 2D Quadrotor (Table 5), and the 3D Quadrotor environment (Table 6). These were obtained through hyperparameter optimization with 50 trials using the tree-structured Parzen estimator of Optuna with the default parameters [1].

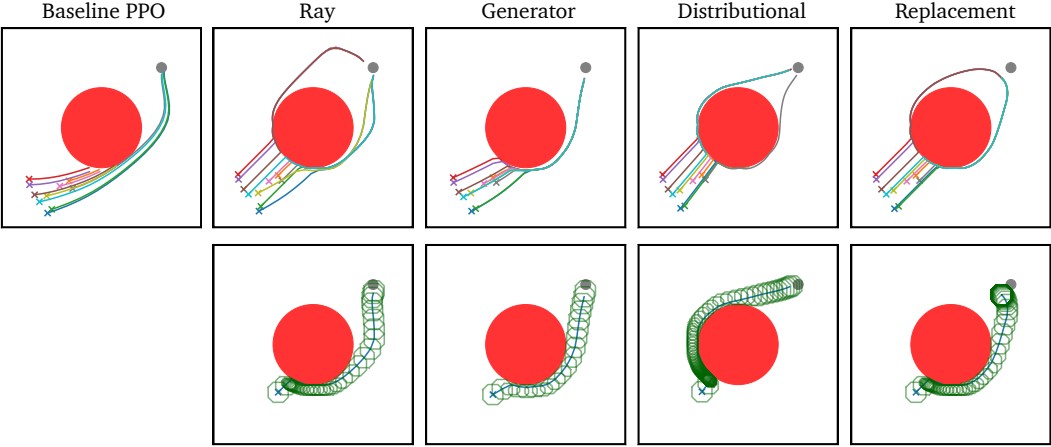

Figure 5: Qualitative deployment results for ten initial states and one goal-obstacle configuration for the Seeker environment. The top half shows ten trajectories with randomly sampled starting states. The lower half depicts the relevant action set (green polygon) for each time step along one trajectory.

Table 4: PPO hyperparameters for the Seeker environment.

| Parameter | Baseline | Ray | Generator | Distributional | Replacement |
|---|---|---|---|---|---|
| Learning rate | $5.43E-5$ | $8.25E-4$ | $3.45E-4$ | $3.85E-5$ | $1.92E-6$ |
| Discount factor $\gamma$ | 0.98 | 0.98 | 0.98 | 0.98 | 0.98 |
| Steps per update | 32 | 256 | 2084 | 32 | 128 |
| Optimization epochs | 4 | 8 | 16 | 4 | 4 |
| Minibatch size | 8 | 128 | 256 | 8 | 128 |
| Max gradient norm | 0.9 | 0.9 | 0.9 | 0.9 | 0.9 |
| Entropy coefficient | $4.71E-5$ | $1.66E-7$ | $6.61E-7$ | $3.33E-6$ | $1.83E-7$ |
| Initial log std dev | $-1.183$ | $-0.010$ | $-0.255$ | $-1.213$ | $-1.064$ |
| Value function coeff. | 0.5 | 0.5 | 0.5 | 0.5 | 0.5 |
| Clipping range | 0.1 | 0.1 | 0.1 | 0.1 | 0.1 |
| GAE $\lambda$ | 0.9 | 0.9 | 0.9 | 0.9 | 0.9 |
| Activation function | ReLU | ReLU | ReLU | ReLU | ReLU |
| Hidden layers | 2 | 2 | 2 | 2 | 2 |
| Neurons per layer | 32 | 32 | 32 | 32 | 32 |

Table 5: PPO hyperparameters for the 2D Quadrotor environment.

| Parameter | Baseline | Ray | Generator | Distributional | Replacement |
|---|---|---|---|---|---|
| Learning rate | $1.24E-4$ | $7.92E-4$ | $4.34E-3$ | $3.94E-4$ | $1.13E-4$ |
| Discount factor $\gamma$ | 0.99 | 0.99 | 0.99 | 0.99 | 0.99 |
| Steps per update | 256 | 1024 | 1024 | 1024 | 512 |
| Optimization epochs | 32 | 8 | 8 | 8 | 8 |
| Minibatch size | 64 | 128 | 128 | 128 | 128 |
| Max gradient norm | 0.9 | 0.9 | 0.9 | 0.9 | 0.9 |
| Entropy coefficient | $8.9E-2$ | $5.65E-2$ | $5.08E-2$ | $5.99E-3$ | $5.42E-3$ |
| Initial log std dev | $-0.437$ | $-0.784$ | $-1.251$ | $-1.217$ | $-1.019$ |
| Value function coeff. | 0.5 | 0.5 | 0.5 | 0.5 | 0.5 |
| Clipping range | 0.1 | 0.1 | 0.1 | 0.1 | 0.1 |
| GAE $\lambda$ | 0.95 | 0.95 | 0.95 | 0.95 | 0.95 |
| Activation function | ReLU | ReLU | ReLU | ReLU | ReLU |
| Hidden layers | 2 | 2 | 2 | 2 | 2 |
| Neurons per layer | 256 | 256 | 256 | 256 | 256 |

Table 6: PPO hyperparameters for the 3D Quadrotor environment.

| Parameter | Baseline | Ray | Generator | Distributional | Replacement |
|---|---|---|---|---|---|
| Learning rate | $2.38E-4$ | $1.08E-3$ | $9.24E-5$ | $7.88E-4$ | $6.25E-4$ |
| Discount factor $\gamma$ | 0.98 | 0.98 | 0.98 | 0.98 | 0.98 |
| Steps per update | 32 | 128 | 128 | 64 | 128 |
| Optimization epochs | 8 | 4 | 16 | 4 | 4 |
| Minibatch size | 16 | 32 | 16 | 64 | 64 |
| Max gradient norm | 0.9 | 0.9 | 0.9 | 0.9 | 0.9 |
| Entropy coefficient | $5.85E-5$ | $1.14E-7$ | $3.41E-7$ | $2.75E-6$ | $1.88E-6$ |
| Initial log std dev | $-3.609$ | $-1.793$ | $-1.363$ | $-1.880$ | $-1.582$ |
| Value function coeff. | 0.5 | 0.5 | 0.5 | 0.5 | 0.5 |
| Clipping range | 0.1 | 0.1 | 0.1 | 0.1 | 0.1 |
| GAE $\lambda$ | 0.9 | 0.9 | 0.9 | 0.9 | 0.9 |
| Activation function | ReLU | ReLU | ReLU | ReLU | ReLU |
| Hidden layers | 2 | 2 | 2 | 2 | 2 |
| Neurons per layer | 32 | 32 | 32 | 32 | 32 |

Table 7: PPO hyperparameters for the Walker2D environment.

| Parameter | Baseline | Replacement | Ray mask | Generator mask |
|---|---|---|---|---|
| Learning rate | $6.992E-5$ | $4.700E-3$ | $2.607E-4$ | $1.719E-4$ |
| Discount factor $\gamma$ | 0.99 | 0.99 | 0.99 | 0.99 |
| Steps per update | 2048 | 2048 | 2048 | 2048 |
| Minibatch size | 128 | 64 | 16 | 32 |
| Max gradient norm | 0.603 | 0.336 | 0.156 | 0.152 |
| Entropy coefficient | $6.559E-7$ | $5.960E-6$ | $4.992E-6$ | $7.488E-5$ |
| Clipping range | 0.165 | 0.131 | 0.102 | 0.192 |
| GAE $\lambda$ | 0.970 | 0.944 | 0.919 | 0.957 |
| Value function coefficient | 0.259 | 0.330 | 0.181 | 0.500 |
| Activation function | ReLU | ReLU | ReLU | ReLU |
| Hidden layers | 2 | 2 | 2 | 2 |
| Neurons per layer | 64 | 64 | 64 | 64 |

