# OpenReview forum: "Excluding the Irrelevant: Focusing Reinforcement Learning through Continuous Action Masking"
_NeurIPS.cc/2024/Conference — NeurIPS 2024 poster_

### Official Review · Reviewer_T9Vb · 2024-07-11

**Soundness:** 3
**Presentation:** 3
**Contribution:** 2
**Rating:** 6
**Confidence:** 3

**Summary:**

Learn a state specific mask for actions. Rather than simply a state specific interval, extend the action mask to different convex set representations. Then, derive a policy gradient for each of these masking schemes.  The masking schemes are ray masks, hypercube transform mask and distributional masks. Applies action masking to seeker and quadrotor tasks and shows that this action masking improves performance.

**Strengths:**

The proposed action masking covers a wide range of possible action mask definitions.

The derived policy gradients are relatively straightforward given the definitions of the action boundaries.

The derivations appear to be sound when applied empirically.

**Weaknesses:**

It is not clear how easy it is to recover the action masking criteria, especially under the more complex generator or distributional schemes, and it seems like this would be rare

The experiments are not particularly convincing because they all follow similar control tasks, but it also seems like these are the only tasks for which the action mask could be easily defined.

**Questions:**

It is not clear why related work is not in a separate section, rather than subsection. There does not appear to be a special connection to the introduction.

It isn't obvious that if G is square and non-singular, that this does not restrict the space of possible relevant action sets, since this would ensure that the hypercube space had an invertible, i.e. one to one, mapping between itself and the action distribution. It seems like many to one would be preferred if the space of the zonotope's hypercube was higher dimension than the action set.

**Limitations:**

While noted in the limitations, deriving a method for identifying not only the policy gradient, but a policy from a learned value function is highly relevant for RL, and it is not clear how these restricted action spaces can be applied to Q value calculations, though it seems reasonable to assume it is possible.

Some suggestions in the main work of how the distributional or generator action space restriction could be defined as a function of the dynamics could be relevant since it seems like these functions have to be hand-engineered, and it is not obvious how to do that in domains where the dynamics are less well defined.

---

> ### Author Rebuttal · Authors · 2024-08-07
>
> We sincerely thank you for your valuable feedback on our manuscript, and for highlighting the broad applicability of our proposed action masking approach. We would like to address your concerns and questions in the following.
>
> # Weaknesses
> ## Action masking criteria
> Thank you for sharing your assessment. There can be different notions of relevance, which has implications for required task knowledge. A relevant action set could encode a formal specification, guarantee the stability/safety of a dynamic system, or enforce system constraints involving interdependent actions (see summary rebuttal A2 for more details). For an explicit relevant action set, one can determine if it is convex. If so, our continuous action masking can be applied. If not, the relevant action set could be underapproximated with a convex set (see rebuttal for reviewer PubC, Q1 for more details). We will clarify the notion of relevance in the introduction and extend Sec. 4.3 with more details on application limitations originating from the assumptions.
>
> ## Unconvincing experiments
> We appreciate your suggestion to include a different type of task with a more intuitive relevant action set. We will extend our empirical evaluation with the Walker2D MuJoCo environment. Here, the Walker2D is interpreted as a battery-driven robot and the relevant action set is based on the total power constraint of the battery. Such a constraint encodes the interdependence of multiple action dimensions and cannot easily be enforced with vanilla RL. Please refer to the summary rebuttal for more details on the new experiment (see A3) and more examples for relevant action sets (see A2).
>
>
> # Questions
> ## Related work as separate section
> Thank you for this remark. We agree and will move the related work subsection to a dedicated section.
>
> ## Clarification on the effect of G being square and non-singular
> Thank you for this question. A generator matrix $G$ that is square and non-singular does restrict the possibilities for relevant action sets, e.g., in two dimensions only rectangles and parallelotopes can be represented but not hexagons. We state in the paper that for this special case of an invertible matrix $G$, the policy gradient for the generator mask does not change (see Proposition 5). You correctly note that it is likely that the generator dimensions $P$ are larger than the action dimensions $N$. Yet, there are cases where relevant action sets with $N=P$ are a valid choice. For example, if the relevant action sets are state-dependent interval sets as for the 3D Quadrotor or if there is a linear dependency between the action dimensions. The latter is the case for the 2D Quadrotor where the force at the left and right rotors should be similar to avoid flipping over. Thus, a parallelotope shape would be a valid choice. Note that existing continuous action masking on interval sets would only poorly cover parallelotope relevant action sets as visualized in the rebuttal PDF, Fig. 2. We will reflect this answer in the paper by clarifying the implications of different choices of $G$ in Sec. 3.2.
>
>
> # Limitations
>
> ## Action masking for value-based RL
> Thank you for your comment. To the best of our knowledge, there is no popular deep RL algorithm for Q-learning in continuous action spaces that is not an actor-critic. For discrete action spaces, previous action masking work has demonstrated its effectiveness [4,6,11,22] and continuous relevant action sets could be discretized to make these approaches applicable.
> In our limitations, we comment on TD3 and SAC, which both learn a policy and for which future work is required to determine the theoretical and empirical effects of action masking on the policy.
> Nevertheless, we see that our statement in Line 149 could be misleading:
>
> > Since the size of the output layer of the policy or Q-function network cannot change during the learning process
>
> We will change it to:
>
> > Since the size of the output layer of the policy network cannot change during the learning process
>
> ## Obtaining relevant action sets and relevance of action masking
> Thanks for your comment. We see that our paper lacked a more explicit specification of the notion of action relevance and a clarification of the practical relevance and benefits of the formulation as part of the policy. We will add this in the revised paper and would like to refer you to A1. and A2. of the summary rebuttal for a detailed answer.
>
> ## References:
>
> [4] Feng et al. 2023
>
> [6] Fulton et al. 2018
>
> [11] Huang et al. 2022
>
> [22] Rudolf et al. 2022

---

> > ### Comment · Reviewer_T9Vb · 2024-08-12
> > **Response to Authors**
> >
> > I appreciate the detailed response, especially the added experiments in walker. While the changes improve the paper, I think they do not significantly improve the score since the core questions remain.

---

> > > ### Author Response · Authors · 2024-08-12
> > > **Response to Reviewer**
> > >
> > > Could you please reiterate what your remaining concerns are?
> > > From our perspective, your detailed review helped us to alleviate the two major weaknesses:
> > > > It is not clear how easy it is to recover the action masking criteria, especially under the more complex generator or distributional schemes, and it seems like this would be rare
> > >
> > > In our new experiments, we now provide an example with a realistic constraint (maximal battery power) that we can very easily represent as a zonotope in the action space. We are convinced that such easy-to-implement constraints are not uncommon in continuous action spaces.
> > >
> > >  > The experiments are not particularly convincing because they all follow similar control tasks, but it also seems like these are the only tasks for which the action mask could be easily defined.
> > >
> > > We believe that the mujoco continuous control tasks Walker2D, Ant, etc. are the de-facto standard benchmark in continuous control. From our point of view these new experiments provide more than convincing arguments for the use of continuous action masking.
> > >
> > > We are happy to discuss these points and any other open concerns.

---

### Official Review · Reviewer_Ke6H · 2024-07-12

**Soundness:** 3
**Presentation:** 3
**Contribution:** 3
**Rating:** 5
**Confidence:** 2

**Summary:**

The paper addresses challenges in RL with continuous action spaces, typically defined as interval sets. These spaces often lead to inefficient exploration due to irrelevant actions. The authors propose three continuous action masking methods to focus learning on relevant actions based on current state, improving predictability and suitability for safety-critical applications. They analyze the implications on policy gradients and evaluate performance using PPO across three control tasks. Results show higher final rewards and faster convergence compared to baseline methods without action masking.

**Strengths:**

**Originality**
- The paper presents a unique perspective on action spaces by utilizing the relevance of action utility in tasks to improve performance. Conventional methods are limited to discrete domains (tasks) so applying their methods to continuous environments was interesting to see.

**Significance**
- The proposed approach has practical implications, especially in complex environments where distinguishing between relevant and irrelevant actions is crucial. Regardless of the coverage of baseline, their methods significantly outperform it establishing the state of the art performance.

**Weaknesses:**

**Reinforcement Learning with Policy Gradients (Section 2.1)**
- L84: "r →: S × A" appears incorrect.

**Continuous Action Masking (Section 3)**
- Assumption 1: Clarify the definition of action relevance.

**Ray Mask (Section 3.1)**
- L131: Need proof that g(a) is bijective.


**Generator Mask (Section 3.2)**
- Why is A(s) suddenly state-dependent? Provide motivation and further description.
- In Proposition 2's proof, the matrix multiplication seems infeasible due to mismatched dimensions (C is N x 1 and Ga results in P x 1).


**Experiment (Section 4)**
- Justify the rationale behind the design choices for action relevance in each environment.
- Compare the chosen action relevance approach to other relevant action settings.

**Results (Section 4.2)**
- Why compare to a standard PPO baseline and not to prior relevant works?
- Include qualitative results to validate the proposed methods.

**Questions:**

See weaknesses.

---

> ### Author Rebuttal · Authors · 2024-08-07
>
> We sincerely thank you for your valuable feedback on our manuscript. We are particularly grateful for the acknowledgment of our manuscript's originality and significance.
> In the following, we outline how we incorporated your feedback and clarify open questions.
>
> ## Improvements to the mathematical formulation
> Thanks for pointing out notation improvements. We reworked the preliminaries and methodology sections to improve notational precision. Specifically, we corrected the following points:
>  - $r \rightarrow: \mathcal{S} \times \mathcal{A}$ to $r : \mathcal{S} \times \mathcal{A} \rightarrow \mathbb{R}$ in [L84].
>  - in the ray mask approach: $g(a) = c + \frac{\lambda_{\mathcal{A}}(a)}{\lambda_{\mathcal{A}^r}(a)} (a - c)$ to $g(a) = c + \frac{\lambda_{\mathcal{A}^r}(a)}{\lambda_{\mathcal{A}}(a)} (a - c)$ in [Eq. 6, L126]
>  - in the generator approach
>     - $g : \mathcal{A} \rightarrow \mathcal{A}^r$ to $g : \mathcal{A}^l \rightarrow \mathcal{A}^r$ in [L163]
>     - $G \in \mathbb{R}^{P \times N}$ to $G \in \mathbb{R}^{N \times P}$ in [L154].
>     - Now we have $G \in \mathbb{R}^{N \times P}$, so the dimensions in [Eq. 12, L164] fit correctly.
>
> ## Clarification of the relevant action space
>
> Thank you for highlighting that the notion of action relevance was not properly defined. We will added a formal definition of the relevant action set to the preliminaries:
>   > *We further assume that we can compute a state-dependent relevant action set $\mathcal{A}^r(s) \subseteq \mathcal{A}$, which potentially reduces the action space, based on task knowledge.*
>
> To introduce our notion of relevant actions, we are adding to the first paragraph of the introduction
>  > *Irrelevant actions are actions that are either physically impossible, forbidden due to some formal specification, or evidently counterproductive for solving the task.*
>
> Additionally, we will clarify that only the relevant action set $\mathcal{A}^r(s)$ is state-dependent, the action sets $\mathcal{A}$ and $\mathcal{A}^l$ are not.
> We would like to refer the reviewer to the summary rebuttal A.2 **Computing relevant action sets** for a more in-depth discussion on this topic.
>
> ## Proof of bijectivity of $g(a)$ in the ray mask
>
> **Lemma**: The mapping function $g: \mathcal{A} \rightarrow \mathcal{A}^r$ of the ray mask $g(a) = c + \frac{\lambda_{\mathcal{A}^r}(a)}{\lambda_{\mathcal{A}}(a)} (a - c)$ is bijective.
>
> *Proof*: We prove that $g(a)$ is bijective by showing that it is both injective and surjective.
>
> For any convex set $\mathcal{A}^r$ with center $c$, we can construct a ray from $c$ through any $a^r \in \mathcal{A}^r$ as $r_d(t) = c + d t$, where $d = \frac{a-c}{\Vert a-c \Vert_2}$, and $t \in \left[ 0, t_{max} \right]$ limits the ray to $\mathcal{A}$. Since $\mathcal{A}^r \subseteq \mathcal{A}$, this also holds $\forall a \in \mathcal{A}$. By construction, any two distinct rays only intersect in $c$. For all points on a given ray, the scaling factors $\lambda_\mathcal{A}$ and $\lambda_{\mathcal{A}^r}$ are constants, allowing us to rewrite $g(a)$ for all $a=r_d(t)$ as the linear function:
> $$
> g \left( r_d(t) \right) = c + \frac{\lambda_{\mathcal{A}^r}}{\lambda_{\mathcal{A}}} \left( r_d(t) - c \right).
> $$
> To show that $g(a)$ is injective, i.e. $\forall a_1, a_2 \in \mathcal{A}$, if $a_1 \neq a_2 \implies g(a_1) \neq g(a_2)$, consider the following two cases for $a_1 \neq a_2$.
>
> Case 1: If $a_1$ and $a_2$ are on the same ray, then, $a_1 = r_d(t_1)$ and $a_2 = r_d(t_2)$ with $t_1 \neq t_2$. Since $g \left( r_d(t) \right)$ is linear and thereby monotonic, it follows that $g \left( r_d(t_1) \right) \neq g \left( r_d(t_2) \right)$ and consequently $g(a_1) \neq g(a_2)$.
>
>
> Case 2: Otherwise, $a_1$ and $a_2$ are on different rays and $a_1 \neq a_2 \neq c$, so $g(a_1) \neq g(a_2)$ follows directly as the rays only intersect in $c$.
>
> Thus, $g(a)$ is injective.
>
> For showing $g(a)$ to be surjective, it is sufficient to show that $\forall a^r \in \mathcal{A}^r$, there exists an $a \in \mathcal{A}$, for which $g(a) = a^r$.
>
> Consider the ray $r_d(t)$ passing through $a^r$. We know that $g \left( r_d(0) \right) = g(c) = c$, and $g \left( r_d(t_{max}) \right)$ is the boundary point of $\mathcal{A}^r$ from $c$ in the direction $d$. Moreover, $a^r$ lies on the line segment between $c$ and $g \left( r_d(t_{\max}) \right)$.
> Since $g \left( r_d(t) \right)$ is linear and continuous, according to the intermediate value theorem, $\exists t^* \in \left[ 0, t_{\max} \right]$, for which $g \left( r_d(t^*) \right) = a^r$ for any $a^r \in \mathcal{A}^r$ along the ray $r_d(t)$.
> As we can construct such a ray through any point in $\mathcal{A}^r$, we have shown that for every $a^r \in \mathcal{A}^r$, there exists an $a \in \mathcal{A}$ such that $g(a) = a^r$, thus proving surjectivity.
>
> ## Additional experiments required
> Based on your recommendations, we added further evaluations as discussed in A3. of the summary rebuttal.
> Additionally, we will add an illustration of typical rollouts on the Seeker environment as interpretable qualitative result (see rebuttal PDF, Fig. 4).
>
> ## Rationale behind the design choices for action relevance in our environments
> Thank you for your question regarding the design choices of our relevant action space.
> A key advantage of our masking approaches is that they ensure that only relevant actions are executed and therefore can be used for safety-critical tasks. There are other notions of relevance, which we explore in the summary rebuttal A2.
> Here, we choose experiments where the relevant action set is a safe action set. In particular, a relevant action is one that avoids collision with an unsafe region. For the Seeker environment, the unsafe region is defined by an obstacle (see Fig. 2). For the quadrotor stabilization experiments, we compute a control-invariant set, which acts as relevant state set $\mathcal{S}^r$.
>
> We will clarify in 4.1.1 and 4.1.2 the motivation and computation of the relevant action set.

---

> > ### Comment · Reviewer_Ke6H · 2024-08-08
> > **Response**
> >
> > Thank you for your response and for proving the bijectivity as well as running a new experiment.
> > Unfortunately, I can't seem to find the information about Fig. 4 so could you elaborate on Fig. 4?

---

> ### Author Response · Authors · 2024-08-09
> **Supplementary figures**
>
> Thanks for your question. Fig. 4 shows qualitative deployment behavior for the different RL agents in the Seeker environment. We use the same goal state (gray circle), same obstacle (red circle), and same ten initial states (colored crosses) for the agents for better comparability. The ten trajectories have the same color as their initial state. Note that for the masking approaches the trajectories are almost identical after passing the obstacle and therefore plotted on top of each other.
>
> The masking agents (ray, generator and distributional) reach the goal state for all ten initial states, with the generator being the quickest. In contrast, the PPO baseline agent reaches the goal with four out of the ten initial states and otherwise collides with the obstacle. The qualitative behavior is aligned with the quantitative results in Table 1 of the submitted paper meaning that the generator mask agent performs best and the PPO baseline performs worst with respect to reaching the goal efficiently.
>
> The goal of the visualization in Fig. 4 is to provide an intuition on the trained agents' behavior. We choose the Seeker environment since we believe it is the most intutive to understand and thus also to interpret the qualitative behavior.
>
> Note that there is unfortunately a typo in the subplot titles: distribution should be distributional.

---

> > ### Comment · Reviewer_Ke6H · 2024-08-11
> > **Response**
> >
> > I understand the setup of Fig. 4. However, as a qualitative result, it lacks relevance to action relevance (as you mentioned, this is merely an alternative to the quantitative results in the table). For instance, it’s not clear how this plot demonstrates the role of action relevance in decision-making.

---

> > > ### Author Response · Authors · 2024-08-12
> > > **Answer to Response**
> > >
> > > Thank you for your reply. We are sorry that we misunderstood your review question for qualitative results. Still, we are not entirely sure how to interpret your comment. If we understand you correctly, you are interested in the impact of choosing different relevant action sets on the decision-making of the agent (i.e., the learning process)?
> > >
> > > Let us briefly provide some intuition on this. While the relevant action sets $\mathcal{A}^r$ in our experiments are all collision-avoidance action sets, their average coverage of $\mathcal{A}$ varies between 25%, 28%, and 70% for the 3D Quadrotor, 2D Quadrotor, and Seeker, respectively. It seems that there is a sweet spot for the restrictiveness of the relevant action set due to two opposing mechanisms. Sample efficiency gains from masking increase, the smaller the relevant action set in relation to the global action set, whereas the exploration capability of the agent decreases. In other words, if the relevant action is almost the entire action space, the sample efficiency gains are likely small compared to not using action masking. If the relevant action set gets too small, the agent cannot learn much due to the very limited amount of possible actions. For example, in our experiments, the reward gain in the Seeker environment is much higher than in the Quadrotor environments, which underscores less constrained exploration. In the extreme case of $\{ a^r \} = \mathcal{A}^r$ always, there is no meaning in using reinforcement learning since the agent cannot explore. Future work will need to investigate these mechanisms in more detail for a multitude of tasks and relevant action sets to provide clearer and more nuanced insights. Yet, we are happy to extend our discussion in Sec. 4.3 (Limitations) in the last paragraph with this intuition.
> > >
> > > If you are interested in a visual demonstration of the relevant action set during a training episode, we can refer you to Fig. 2 of our original paper. It shows the relevant action set at a certain state in the Seeker environment, which prohibits the agent from colliding with the red obstacle.

---

> > > > ### Comment · Reviewer_Ke6H · 2024-08-13
> > > > **Response**
> > > >
> > > > So, is the relevant action set always provided to the agent, rather than the agent (e.g., the generator generating the latent action set) computing it on its own like the relevant action set being state dependent (A(s))? I wanted to observe how the relevant action set varies during the episode in relation to the agent's decision-making. When the agent performs poorly, this action relevance set fails to form a meaningful polygon in the entire action space.

---

> > > > > ### Author Response · Authors · 2024-08-13
> > > > > **Anser to Response**
> > > > >
> > > > > Thank you for the clarification. The relevant action set is state-dependent and, for our experiments, computed with the optimization problems specified in Eq. 35 and 38 in the Appendix. Generally, we assume that $\mathcal{A}(s)$ is computed or derived by another algorithm, which is independent of the RL agent policy (see summary rebuttal **A2. Computing relevant action sets** for more details). For our experiments, the relevant action sets become smaller the closer the agent navigates to the obstacle (Seeker) or boundary of the relevant state set $\mathcal{S}^r$ (Quadrotors). For the Quadrotor environments, the task is to stabilize at the center of $\mathcal{S}^r$. Thus, with the policy improving, the agent chooses fewer actions that lead to moving toward the boundary of the relevant state set $\mathcal{S}^r$. For the Seeker environment, the goal is to reach a position while the direct path is blocked by an obstacle. Therefore, navigating close to the obstacle is optimal, and the agent actively navigates to states with smaller relevant action sets. More specifically, using the zonotope with 4 generators (as depicted in Fig 2), the relevant action set covers approximately 87.5% of the action space when the obstacle is too far away to collide in the next time step. When the agent is at the border of the obstacle it reduces to about 40%. To put these numbers in perspective, the relevant action set coverage for the example state in Fig. 2 is 44%.
> > > > >
> > > > > To make the above intuition clearer, we can add an extended version of Fig. 4 of the rebuttal PDF to the Appendix of our revised paper. The extension would be a subplot depicting the changes in the relevant action set coverage of the action space along the trajectories. Additionally, we can provide a plot that shows how the average relevant action set coverage changes over the training process in the Appendix of the revised paper.

---

### Official Review · Reviewer_PubC · 2024-07-13

**Soundness:** 3
**Presentation:** 4
**Contribution:** 4
**Rating:** 6
**Confidence:** 4

**Summary:**

This paper discusses methods for action masking in continuous action spaces to improve convergence stability and sample efficiency in reinforcement learning. The paper introduces three methods for action masking with convex relevant action sets, proves their convergence, and experimentally verifies their effects.

**Strengths:**

This paper is excellently written, defines a clear and well-motivated goal, and describes three intuitive and theoretically grounded methods to achieve that goal within well-defined and clearly stated limitations.

To my knowledge these approaches are novel (though the distributional mask I suspect has been used as a one-off solution in prior work as it is conceptually very simple), and their definition and analysis are nontrivial.

**Weaknesses:**

This paper is pretty solid overall, and I have few major complaints.

The one significant issue I see is that I think the distributional mask algorithm is off-policy by nature, meaning it's use with on-policy methods like PPO is biased and will cause performance loss or divergence. This may explain the observed underperformance of this masking method in two of the three experimental tasks, and while the algorithm can clearly converge in some cases it seems like a major issue with that particular mask in the context of PPO (off-policy algorithms could use it without issue, but those are left to future work here) that should be noted, or the mask omitted from this paper and left to future off-policy methods.

Beyond that, the experimental evaluation is relatively simple (though I think it is sufficient to validate these algorithms), and more challenging tasks would be useful to demonstrate the limitations of these masking methods. That said, the paper makes it clear that defining a suitable convex relevant action set is a manual process and can be challenging (this is okay as a limitation), so it is understandable why such stress tests are not performed. If there was a way to increase difficulty without major manual action set definition work it would strengthen the evaluation to include it.

I have a few other minor issues and questions noted below, but overall this is a paper that is clear in its goals and describes methods that achieve them, validated to a reasonable standard of theoretical and experimental evidence. There's more that could be done on this topic, but the contribution of this paper is significant on its own, so I'm inclined to recommend acceptance (particularly if something is done to address my concern about the distributional mask above).

**Questions:**

-Is assumption 1 (relevant action set is convex) reasonable in most cases? I can imagine disjoint action subsets being relevant in many cases- for example, a self-driving car that needs to steer either left or right to avoid collision, but not go straight ahead or backwards.

-I'm not sure it's actually necessary to compute the policy gradient across the action mask (with the exception of the distributional mask). Once an action is sampled from the policy, the mapping to the relevant action set can simply be treated as part of the environment/transition function which the policy can learn to manipulate without gradient flow. Does this simplify things or am I missing something? This would also permit arbitrary nonconvex relevant sets, I believe.

-For the gradient of the distributional mask in proposition 4, isn't this affected by off-policyness due to the constrained sampling of actions from the policy distribution? For example, if most of the policy distribution probability mass lies outside the relevant set (e.g. in the event of transfer learning to a different task with a new relevant set) the actions sampled will not follow the policy distribution closely and thus \pi_{\theta}(a|s) will not be an accurate probability of sampling action a at state s. As noted above, this seems like a big issue that should be noted or addressed, unless I'm missing something that corrects for the off-policyness.

-Small quibble: The goal in figure 2 looks black to me on two different monitors, perhaps using a lighter gray would make it more distinct from the agent position?

-It would probably be reasonable to move the environment definitions for the two quadrotor tasks to the appendix to save space in the main paper, FWIW. I'm not sure the abbreviated version that's present provides all that much context over the text descriptions of the tasks.

-It's not critical to have, and I realize it's a difficult thing to derive a relevant set for by nature, but having an experiment on an environment with 10+ action dimensions would be a nice addition to demonstrate that these masking approach can scale to higher dimensional action spaces tractably. I'd also appreciate some comments on compute cost scaling with action dimension count in the limitations or conclusions sections, if possible, since it seems like compute cost is likely to increase with the dimensionality of the action space.

**Limitations:**

This paper does an excellent job of making clear its limitations and scope.

I don't see any potential negative societal impacts from this work.

---

> ### Author Rebuttal · Authors · 2024-08-07
>
> Thank you for your insightful comments and for recognizing the novelty and theoretical grounding of our proposed methods. In the following, we respond to the weaknesses (W1 and W2) stated and questions (Q1 - Q6) raised.
>
> # Weaknesses
>
> ## W1. Distributional mask being off-policy by nature
> Thank you for sharing your reading of our distributional mask approach. Could you reiterate on why you would refer to the distributional mask approach as off-policy? We would refer to it as on-policy as we are directly sampling from the relevant policy $\pi_\theta^r(a^r|s)$, which is normalized through the integral $\int_{\mathcal{A}^r} \pi_\theta(a | s) \mathrm{d} a$, and derive the corresponding gradient $\nabla_\theta \log \pi_\theta^r(a^r | s)$. However, our implementation approximates the gradient by treating the integral as a constant, which introduces some off-policyness. However, such off-policyness is also introduced by most stochastic on-policy RL implementations (including PPO) through the clipping of samples from the normal distribution.
>
> ## W2. Simplicity of experiments
> Thank you for your assessment. The main goal of our experiments is to provide initial empirical evidence for continuous action masking and compare the three masking approaches. To strengthen our evaluation, we will add a Walker2D Mujoco environment and an additional baseline to the revised paper as described in A3. of the summary rebuttal and 3. in rebuttal for reviewer 8567.
>
> # Questions
>
> ## Q1. Convex set assumption
> Thanks for your question.
> Note that convex sets are an important generalization of current practice that allows for interval sets only.
> We believe that assuming relevant action sets are convex is reasonable since exclusion of extreme actions and dependencies between action dimensions can be described by them. Additionally, one can underapproximate a non-convex relevant set with a convex relevant set at the cost of excluding some relevant actions [https://arxiv.org/abs/1909.01778].
> For disjoint convex sets, our continuous action masking could be extended with a discrete action that represents a decision variable switching between the disjoint subsets. To find an optimal policy, hybrid RL algorithms could be used [https://arxiv.org/abs/2001.00449]. Yet, this major extension of continuous action masking is subject to future work, and we will add these considerations in Sec. 4.3.
>
> ## Q2. Mapping as part of the environment
> Thank you for your question. Formulating action masking as part of the policy leads to more intuitive action spaces, potential integration of the masking map in the gradient, and more possible masking approaches (see summary rebuttal, A1). Note that implementing the ray mask on the policy or environment side is mathematically identical due to the unchanged gradient. To assess the empirical relevance of our formulation, we added another baseline where as part of the environment dynamics irrelevant actions are replaced with relevant ones. Please refer to 3. in rebuttal for reviewer 8567 for more details.
>
> ## Q3. Gradient of the distributional mask
> You are correct in your assessment that $\pi_\theta(a | s)$ is not an accurate probability of sampling action $a \in \mathcal{A}^r$ at state $s$. To address this issue, we normalize the constraint distribution with the integral in equation (15) and use the probability of the resulting distribution $\pi_\theta^r(a | s)$ for sampling and in the objective function.  With this formulation, the entire probability mass of the resulting distribution $\pi_\theta^r(a | s)$ lies within $\mathcal{A}^r$, and the distribution accurately reflects the actual likelihood of sampling action $a^r$.
>
> ## Q4. Goal color
> Thanks for the hint. We will change the color in the revised paper.
>
> ## Q5. Dynamics to the appendix
> Thanks for the suggestion. We will extend Sec. 4 with the additional experiments as described in our summary rebuttal and move the details of the dynamics to the appendix.
>
> ## Q6. Experiment with higher action dimensions
> Thank you for your suggestion to extend our evaluation. To provide empirical insights for higher dimensional action spaces, we will add the Walker2D Mujoco environment as described in the summary rebuttal (see A3). Although the Walker2D has less than ten action dimensions, it has the highest action dimensionality of the four environments with six dimensions.
>
> The compute costs are mainly affected by the calculation of the relevant action set in each state and by the additional operations for the mapping function $\mathcal{F}$ of the specific masking approach.
> The former depends strongly on the notion of relevance. For our experiments with collision avoidance as notion of relevance, we compute the relevant action sets at each state with an exponential cone program, which scales polynomially with system dimensions (i.e., state and action) assuming a suitable interior-point solver [Boyd, 04, Convex Optimization].
> The second aspect important for the compute costs is the specific masking approach.
> For the ray mask, the cost is dominated by computing the boundary points, which is a linear program for zonotopes with polynomial complexity in the state dimension [Kulmburg, 21, On the co-NP-Completeness of the Zonotope Containment Problem]. For the generator mask, the matrix multiplication of $G$ with $a^l$ is the dominating operation. For the distributional mask, sampling with the random-direction hit-and-run algorithm introduces computational cost. With zonotopes, each step involves solving two linear programs to determine the boundary points. As mentioned in line 190 the mixing time, or the number of steps before accepting a sample, is set to $N^3$, where $N$ is the action dimension.
>
> We will clarify the computational cost scaling in Sec. 4.3 and extend our runtime table in Appendix A.6 with the Walker2D environment, for which the PPO baseline and the generator mask training run at the same speed.

---

> > ### Comment · Reviewer_PubC · 2024-08-12
> > **Response to Rebuttal**
> >
> > Thanks for your detailed comments.
> >
> > -Regarding the off-policyness of the distributional mask, consider the case where the policy is a unit normal distribution and the mask is a truncated unit normal (which I believe is consistent with section 3.3). Actions sampled from this truncated distribution will have a different probability than they would under the original unit normal (e.g. because some actions are excluded, those that are included have a higher probability than otherwise). The exact gradient from Proposition 4 cannot be computed (as noted in lines 198-200), so this algorithm is fundamentally off-policy since it's impossible for it to be on-policy using the truncated distributions discussed in section 3.3 (maybe there's other distributions that would avoid this issue but those aren't discussed and seem like a non-trivial extension to define).
> >
> > Further, I disagree with the assumption that the difference between the truncated and original distributions will be trivial in practice, both because it can be hard to estimate where policy gradients will push the distribution in complex tasks with limited samples per gradient step and because just how truncated the distribution is depends greatly on the task- some tasks could require very aggressive truncation to limit to only the relevant set, while others might be more permissive. The other masks seem fine to me, but this one seems flawed in a way that is hard to solve and will matter in practice as well as theory.
> >
> > I'm not sure what "the clipping of samples from the normal distribution" refers to for PPO/etc. In every PPO/A2C/etc implementation I've worked with the full distribution gets sampled. Can you clarify?
> >
> > Given this, I'd strongly recommend removing the distributional mask method from the paper, or at least making it's off-policyness clear. I'm certainly willing to accept experimental evidence that it works ok in practice, but the experiments presented here don't seem all that reassuring and don't motivate why a theoretically flawed algorithm might be superior in practice (it doesn't outperform the other masking algorithms in the best case and seems to underperform sometimes).
> >
> >
> > -Regarding handling non-convex relevant action sets, I'm sorry but I find the assertion that a convex subset of the full non-convex relevant set is good enough to be unconvincing. When looking at robotics (one of, if not the, biggest application domain for continuous-action RL) one finds non-convex (particularly disjoint- go left or right, but not straight ahead, etc) relevant sets all over the place. I'd need experimental evidence to believe this isn't an issue in practice, but I'm willing to accept this as a topic for future work (not every limitation of an algorithm needs to be solved in one pass). It does seem worth noting as a limitation (sounds like this will be added) and is a mark in favor of environment-masks over policy-masking that it supports disjoint sets by default, however.
> >
> > Considering the above and the discussion/comments of other reviewers, I think I shall maintain my score. I do think this paper could be made stronger, but it's also reasonable to leave these questions to future work.

---

> > > ### Author Response · Authors · 2024-08-12
> > > **Answer to Response**
> > >
> > > > The exact gradient from Proposition 4 cannot be computed (as noted in lines 198-200), so this algorithm is fundamentally off-policy since it's impossible for it to be on-policy using the truncated distributions discussed in section 3.3.
> > >
> > > You are correct by stating that if one cannot compute the gradient, the distributional mask becomes off-policy, since we update the parameters for a different distribution. However the fact that the integral is intractable does not imply that it cannot be approximated or derived in a different way (granted, our wording of "Since we cannot compute the gradient..." is suboptimal). However, this is certainly not trivial and likely warrants considerable future work.
> > >
> > > > Further, I disagree with the assumption that the difference between the truncated and original distributions will be trivial ... while others might be more permissive.
> > >
> > > Saying that the difference is trivial in practice is certainly an understatement, but we believe that we did not make this statement anywhere. We agree that estimating the practical implications is challenging. Our suggestion that approximating the gradient might be valid in practice stems from the theoretical analysis presented in the "Physical implications of ... (2/2)" section of the general discussion. By approximating the gradient, we remove the option 2. for the gradient to increase the log-likelihood of the sample. Since this option can only favor actions close to the boundary of the relevant action set, it might not be applicable often in practice.
> > >
> > > > In every PPO/A2C/etc implementation I've worked with the full distribution gets sampled. Can you clarify?
> > >
> > > We acknowledge that our previous answer did not clearly state this point. In nearly all continuous reinforcement learning tasks, sampled actions are clipped to specific ranges (typically $[-1, 1]$) due to a fixed interval action space usually based on physical constraints. This clipping introduces a bias when using a Gaussian distribution as the policy, as its infinite support creates boundary effects. Chou et al. (2017) provide a thorough analysis of this issue in their paper "Improving Stochastic Policy ... " (Section 3.2).
> > > Essentially the clipping creates a form of truncated distribution with spikes at the truncation limits. One could argue that neglecting this effect inherently makes most stochastic policy gradient methods off-policy (note that we don't claim this is necessary the case!). PPO does not account for this effect, yet it demonstrates strong performance in practice. Based on this, we infer that our gradient approximation might also have little impact on performance.
> > >
> > > > Given this, I'd strongly recommend removing the distributional mask method from the paper, or at least making it's off-policyness clear.
> > >
> > > We believe it's important to include the distributional mask in the paper despite its current limitations. This method employs a novel technique for sampling from the policy distribution, which is rarely used in RL and thus represents a contribution. By publishing this work, even with its current need for gradient approximation, we open the door for other researchers to build upon and potentially solve the existing problems. We recognize that for this justification it is crucial to explicitly state the current limitations of the method, including its off-policy nature.
> > >
> > > > Regarding handling non-convex relevant action sets, I'm sorry but I find the assertion that a convex subset of the full non-convex relevant set is good enough to be unconvincing.
> > >
> > > We appreciate your concern about non-convex relevant action sets, particularly in robotics applications. We want to clarify that we didn't intend to suggest that convex sets are universally "good enough". Rather, they represent a significant improvement over the current practice of using only intervals as sets.
> > > We fully acknowledge that for many applications, especially in robotics, restricting actions to convex sets can be an oversimplification, as Your example illustrates.
> > > Our current work with convex sets is a step towards more flexible action spaces, but we agree that handling non-convex sets would be a valuable extension of our methods. However, this presents significant challenges that we believe warrant separate, focused research.
> > >
> > > > It does seem worth noting as a limitation (sounds like this will be added) and is a mark in favor of environment-masks over policy-masking that it supports disjoint sets by default, however.
> > >
> > > We don't really understand why you are certain that environment-masks support convex sets by default. None of our methods can be applied to non-convex sets as is, even if they are implemented in the environment. Can you please elaborate how this would be done?  We can imagine that action replacement is implemented with non-convex sets. Yet, this is conceptually different to masking and the newly added action replacement baseline performs worse than masking (see rebuttal for reviewer 8567, 3.).

---

> > > > ### Comment · Reviewer_PubC · 2024-08-14
> > > > **Response to Response**
> > > >
> > > > Regarding gradient approximation/offpolicyness issues, I'll simply say that while yes an approximation can be useful, it needs to be an unbiased approximation, or if biased that bias needs to be empirically trivial. Given that the distribution mask underperforms the others in some test cases, I'm not sure that's good evidence for the bias being trivial in practice.
> > > >
> > > > Regarding environment masking supporting non-convex sets, removing the need to compute a gradient means non-differentiable functions can be used (for example, mapping a continuous action space from -1 to 1 to two disjoint sets, -1 to -0.5 and 0.5 to 1). The shape of the masking function is then only limited by the ability of the designer to implement it depending on the task, since the policy can learn to interact with it regardless.

---

> > > > > ### Author Response · Authors · 2024-08-14
> > > > > **Answer to Response**
> > > > >
> > > > > Thank you for your reply and clarifications.
> > > > >
> > > > > > Regarding gradient approximation/offpolicyness issues, I'll simply say that while yes an approximation can be useful, it needs to be an unbiased approximation, or if biased that bias needs to be empirically trivial. Given that the distribution mask underperforms the others in some test cases, I'm not sure that's good evidence for the bias being trivial in practice.
> > > > >
> > > > > We agree that the approximation introduces bias and will highlight this more prominently in the limitations. Yet, we believe that the theoretic introduction and empirical evaluation of the distributional mask add value to our paper. It shows that naively translating discrete action masking to continuous action masking creates a challenge for calculating the gradient and is not the best-performing method for our experiments. The empirical impact of the bias introduced by the approximation of the gradient was a priori unclear and could have been compensated by maintaining the shape of the policy distribution within the relevant action set. This certainly merits further methodological and empirical investigation.
> > > > >
> > > > > > Regarding environment masking supporting non-convex sets, removing the need to compute a gradient means non-differentiable functions can be used (for example, mapping a continuous action space from -1 to 1 to two disjoint sets, -1 to -0.5 and 0.5 to 1). The shape of the masking function is then only limited by the ability of the designer to implement it depending on the task since the policy can learn to interact with it regardless.
> > > > >
> > > > > Thank you for clarifying. If the relevant action set is not state-dependent, it might be relatively easy to design such a mapping of disjoint sets in the environment. Yet, when the relevant action set is state-dependent and computed online, it is likely to be impossible to foresee all kinds of relevant action sets and, thus, also difficult to design a mapping function. Therefore, we believe that in such cases action replacement would be used, for which only a function that replaces irrelevant with relevant actions needs to be engineered. To reiterate, our framework also allows for non-differentiable functions, if the function is bijective (such as the function in your example).

---

### Official Review · Reviewer_8567 · 2024-07-15

**Soundness:** 3
**Presentation:** 3
**Contribution:** 2
**Rating:** 6
**Confidence:** 4

**Summary:**

This paper proposes mathematical formulations for continuous action masking in reinforcement learning, to incorporate domain-knowledge in the form of state-specific sets of relevant actions. It introduces 3 functional forms to extract relevant actions from the original action space, and consider its effect on the policy gradient. The policy gradient does not change much, and the paper shows that the forms compare similarly and better than learning an agent without any knowledge of the continuous action mask at all.

**Strengths:**

- The problem of action masking in continuous action space is an underexplored one, but could have major impact in efficacy of agents and incorporating domain-specific knowledge.
- The proposed continuous action masking could potentially be useful for safety demarcations.
- The paper provides mathematical frameworks to formulate continuous action masking and also derive their (minimal) effect on the policy gradient.
- The paper is mostly well-written and explains the mathematical derivations quite well. Quick note that Section 2.1 and 2.2 could be made more integrated, currently they seem completely disconnected.

**Weaknesses:**

## 1. General applicability of this paper's ideas
Obtaining a state-specific relevant action set can be really hard. The paper, however, makes contradictory statements about this:
- L5: "little task knowledge can be sufficient to identify significantly smaller state-specific sets of relevant actions."
- L284-285: "assume that an appropriate relevant action set can be obtained. Yet, obtaining this can be a major challenge in practice."

From the experiments on the 3 environments, it already seems like defining the relevant action set requires a lot of domain knowledge about the state space features and the dynamics function.

As of now, there does not seem to be any way the ideas in this paper could be useful for any practical domain.
- Can the authors provide some concrete examples of how one can obtain such relevant action sets for problems of practical interest and scale?
- Can the authors provide any results on a commonly used continuous action space RL benchmark?

## 2. Gains not coming from the policy gradient, but only because of constraining the action space
The paper's proposed formulation is interesting because it uses continuous action masking as part of the learning policy and informs the policy gradient update about the continuous action mask. However, when we look at the resultant policy gradients for each mask in Eq. 10, Line173, and Line199, it seems that the policy gradient simply reduces to $\lambda_\theta log \pi_\theta(a | s)$ for all cases.

So, the effective change in implementation is just how the action to take in the environment is model: $a^r = g(a)$. But, this **doesn't utilize continuous action masking to improve the policy learning objective** in any meaningful way. Is my understanding correct in this?

Another observation that validates the claim that policy learning is not influenced much is seen from the results and L248-249. The initial rewards themselves are significantly higher, which means that the action mask just reduces the space of exploration of the agent so much that, as long as it takes a valid action, it would get a high reward.

## 3. Simpler baselines for continuous action masking
Continuing from the above point, if all that needs to be done is to compare different formulations of g(a), there is a much simpler alternative perspective:
- Action-Masking as part of environment: Simply, apply the action mask as part of the environment, without changing the PPO objective at all. So, there is an action-masking filter before executing an agent's action in the environment, and ignore if the action is invalid.
- Sampling-augmented action-masking: Keep sampling actions from $\pi_\theta$ until you find a valid action that can pass through the known continuous action-masking map.

The current PPO baseline is very weak, and does not utilize action-masking at all. It seems most of the learning in PPO is going into the effort of learning the action mask. To really justify this paper's proposed action masking schemes are useful, they must compare against other forms of naive action masking, including the two listed above. This perspective on considering the action mask as part of the environment is also much more generally applicable and does not require any change to the policy gradient update.

**Questions:**

Listed in weaknesses.

**Limitations:**

The authors discuss several limitations of the work, but the important ones of applicability and baselines need to be addressed.

---

> ### Author Rebuttal · Authors · 2024-08-07
>
> We thank you for your thoughtful and critical comments which helped us to strengthen our arguments for the utility of action masking. We address your questions below.
>
> ## 1. General applicability of this paper's ideas
> Thank you for pointing this out. We agree that the two statements appear contradictory at first glance. However, "can" removes the contradiction in our view. In line 5, “can be” indicates that in some cases, minimal task knowledge might be enough to identify relevant actions. Using "can be obtained" in lines 284-285 highlights that while it is possible to obtain a relevant action set, it may also be quite challenging in many practical situations. Thus, our seemingly contradictory statements highlight the spectrum of difficulties to obtaining a relevant action set, on which we elaborate further in the summary rebuttal (A2). We will highlight this spectrum better in the revised version of this paper.
>
> As we mention in the paper, action masking can be especially useful for safety-critical applications, where relevance means collision-avoidance. Regarding your first point, we provided such concrete examples for the application of autonomous driving in A2 of the summary rebuttal. To address the second point, we conducted an additional experiment for the MuJoCo Walker2D task, which we justify and evaluate in A3 of the summary rebuttal.
>
> ## 2. Gains not coming from the policy gradient, but only because of constraining the action space
> Thanks for your question. We defined action masking as transforming the unmasked policy $a \sim \pi_\theta(a | s)$ to the masked policy $a^r \sim \pi_\theta^r(a^r | s)$, for which $a^r \in \mathcal{A}^r$ always holds. While most stochastic policy gradient algorithms utilize normal distributions with well-known policy gradients $\nabla_\theta \log \pi$, masked policies generally do not follow standard normal distributions. Consequently, we derived gradients for these masked distributions to provide a mathematically sound method. It's important to note that we do not claim these adapted gradients necessarily improve algorithm performance in general.
>
> Nevertheless, it is not true that all masked policy gradients always reduce to the unmasked policy gradient $\nabla_\theta \log \pi_\theta(a | s)$. This only applies to the ray approach, and the proof for it is not trivial, making it also an important contribution. For the generator mask, the policy gradient remains the same, only if the generator matrix is invertible (line 173), which occurs solely when the zonotope has $2^N$ vertices. The distributional mask yields a substantially different gradient (see equation (17)). In practice, we approximate it with the original gradient, since the integral over $\mathcal{A}^r$ is intractable. We acknowledge this simplification in Sec. 4.3, and suggest potential approximations for future work.
>
> Our primary intention was not to claim that the derived gradients universally improve RL algorithm performance. Rather, we emphasize that deriving correct gradients for modified distributions is crucial for establishing a mathematically sound method, thus forming an essential part of our contribution.
>
> Regarding your second observation, we agree that most of the learning of vanilla PPO goes into the effort of learning to select relevant actions. However, we see this as a justification for the utility of action masking because it allows us to encode task knowledge directly into the policy and, thereby reducing the exploration space. Nevertheless, we do not intend to claim that action masking is inherently better than standard PPO, which we will make sure to clarify in our revised paper. Yet, our experiments do show that utilizing relevant action sets with action masking can drastically improve the convergence speed, albeit at a higher computational cost.
>
> ## 3. Simpler baselines for continuous action masking
> Thank you for proposing two additional baselines for comparison. However, we do think that our implementation and experiments already cover both of them.
>
> The ray mask is a version of the first proposed baseline. As derived for proposition 1, the ray mask does not affect the gradient of the policy distribution (because $g(a)$ is bijective), which makes it mathematically equivalent to being defined as part of the environment. We further address the proposal of generally viewing action masking as part of the environment in A1 of the summary rebuttal.
>
> The second proposed baseline is mathematically equivalent to our distributional mask, since it only modifies the sampling procedure to use rejection sampling instead of geometric random walks. We initially used rejection sampling, but quickly found that it carries a significant downside, which makes it not applicable to RL. Consider the case in two dimensions, where $\mathcal{A}^r$ is a small set centered at $\left[ -0.5, -0.5 \right]^T$. If the policy $\pi_\theta(a | s)$ defines a normal distribution with mean at $\left[ 1.0, 1.0 \right]^T$ and small variance, the likelihood of a sample $a \sim \pi_\theta(a | s)$ being inside $\mathcal{A}^r$ is almost zero. This can cause the algorithm to get stuck at a sampling step, which we observed in practice. The issue even aggravates in higher dimensions due to the curse of dimensionality.
>
> Nevertheless, we acknowledge that the PPO baseline may be considered relatively weak. We add a comparison to another common approach for adapting the actions of RL agents; action replacement [14]. This method substitutes actions outside the relevant action set with randomly sampled actions from within it. We depict the comparison in Fig. 1 of the rebuttal PDF. In the experiment, masking performs as good as action replacement in the 2D Quadrotor task, better in the 3D Quadrotor, and significantly better in the Seeker environment. These result suggests that action masking is a competitive approach with respect to a baseline that also explores the relevant action set only.
>
> [14] Krasowski et al. 2023

---

### Author Rebuttal · Authors · 2024-08-07

Dear reviewers,

Thank you for your thoughtful comments and questions. We address general points below.

## A1. Relevance of continuous action masking as part of the policy

Action masking enforces task knowledge by focusing learning on relevant actions, thereby increasing sample efficiency and reducing the need for reward tuning - two common practical problems in RL. The existing literature mostly regards discrete action spaces and shows masking is highly effective [4,6,11,22]. However, real-world systems often operate in continuous action spaces, for which only interval action masks have been explored [14]. We develop three methods that enable action masking for arbitrary convex sets. With our experiments, we demonstrate that continuous action masking can increase convergence speed and performance significantly.

While reviewers 8567 and PubC suggest to define action masking as part of the environment, we argue for incorporating it into the policy distribution, as done for discrete action spaces [11]. This has three main advantages. First, this formulation is more intuitive since the relevant actions $a^r$ stay interpretable in the original action space $\mathcal{A}$. E.g., for the generator mask, adding the mapping function $g(a)$ to the environment would result in an action space with a dimension for each generator of the relevant action zonotope. An action would be the generator factors, for which the real-world meaning is not intuitive. Second, our formulation allows to incorporate information about the modification into the gradient. This is relevant for the generator and distributional mask where the relevant policy gradient is different to the original policy gradient (see Proposition 3 and 4). Since computing relevant action sets can be costly, it is desirable to use this information in the backward pass as well. Third, action masking as part of the policy can be used for more formulations. Specifically, the distributional mask cannot be moved to the environment since it directly modifies the sampling of the policy distribution. We will clarify these advantages in Sec. 3 of the revised paper.

## A2. Computing relevant action sets

We assume that a set of relevant actions is computable. A relevant action set reduces the action space, based on task knowledge. This is a flexible definition that allows for different levels of complexity and required task knowledge. Thus, action relevance is a spectrum and the specific definition is a design choice. For our experiments, we equate relevance with guaranteed collision-avoidance since this is an important feature for safety-critical tasks that standard RL does not achieve [14]. For this highly interpretable and strict notion of relevance, the necessary level of task knowledge (e.g., system dynamics and unsafe sets) and computational cost is high. Yet, this can be a reasonable effort for gaining safety guarantees.

However, there are many other notions of relevance with substantially different levels of required task knowledge and compute. Let us illustrate this for autonomous driving. Large steering angles at high velocity potentially destabilize a vehicle, i.e., can be seen as irrelevant actions. Relevant action sets that restrict steering angles depending on the velocity are trivial to derive. Another notion of relevance is compliance with traffic rules [R1], e.g., only turning right in a right-turn lane, or not accelerating in front of a red light. Here, a medium amount of task knowledge (e.g., road map) is required and the relevant action sets for a single rule are still straightforward to compute (e.g., only allowing steering that leads to a right turn). Similar to our experiments, one can also define relevance as collision avoidance for driving. Due to a highly dynamic environment with other participants, this leads to high required task knowledge. Yet, this is not infeasible as demonstrated by [R2]. Note that there is also recent work on obtaining relevant action sets in a data-driven manner [R3].

Nevertheless, we agree that action relevance and resulting implications for computational costs are not described sufficiently, hence we will clarify this in the revised paper.

## A3. Additional experiments

While we believe that our experiments provide sufficient initial evidence validating our theoretical contributions to action masking, we agree with the reviewers that an evaluation of our continuous action masking approaches on a more diverse set of tasks would further strengthen our claims. Therefore, we are adding experiments on the Walker2D environments (https://gymnasium.farama.org/environments/mujoco/walker2d/). Our concept here is to define the relevant action space as all actions for which $||a||_1 \leq \alpha_p$. This can be viewed as a maximal power output constraint $\alpha_p$.

The results of the experiment are depicted in Fig. 3 of the rebuttal PDF. We observe that the generator and ray mask both learn a performant policy and that the generator mask slightly outperforms standard PPO. Further, encoding the relevant action space as a termination condition prohibits standard PPO to learn. Note that distributional mask is excluded, since it's slow computation time failed to produce results in the available time. In conclusion, this small experiment further highlights the practical utility of action masking.

In addition, we add a comparison to an common approach that adapts irrelevant actions in the environment; action replacement [14]. Our experiment shows that action masking performs better or as good as replacement depending on the environment (see Fig. 1 of rebuttal PDF).

## References:

[4] Feng et al. 2023

[6] Fulton et al. 2018

[11] Huang et al. 2022

[14] Krasowski et al. 2023

[22] Rudolf et al. 2022

Additional:

[R1] Mehdipour et al. 2023. "Formal methods to comply..."

[R2] Wang et al. 2023. "Safe Reinforcement Learning for Automated Vehicles..."

[R3] Theile et al. 2024 "Learning to Generate..."

---

> ### Comment · Reviewer_8567 · 2024-08-08
> **Trying to understand why action masking should be considered part of policy and not environment**
>
> Thank you for your rebuttal response. I am posting here for a potentially joint discussion with other reviewers (like PubC) who shared my question about `treating action masking as part of the environment`.
>
> I want to understand this part better. I still believe the main benefit of formulating action masking as part of the policy comes from how it can help the policy parameters learn about what the action mask is, and thus, help it learn a better policy. For this to happen, there has to be a change in the gradient objective. I have been thinking about this in depth, and have some thoughts / questions.
>
> 1. Let's think intuitively at a high level. Is it even physically possible that gradient information can be passed into the policy? Say, I am a neural network (policy) and my outputs are transformed in a particular way, via $\mathcal{F}$ from $\pi(a|s) \to \pi^r(a^r | s)$, and then it goes through a non-differentiable operation (environment, E) to give a return R. Now, the gradients for me can still only flow through my outputs, i.e., $\pi(a|s)$. Why would separating what happens after my output be any helpful to me, i.e., how can I extract some information by separately treating $\mathcal{F}$ and E? Even if I knew what $\mathcal{F}$ is, I don't know what E is, and I can only see the final effect in the form of the return R. Is the knowledge of $\mathcal{F}$ adding any extra information above the knowledge of return R, or is my gradient conditionally independent of $\mathcal{F}$ given R?
>
> 2. For the case when $a^r = g(a)$ and g is bijective, we know that the gradient of $\pi^r$ = gradient of $\pi$. Then, why do we believe that when g is not bijective there will be any extra information added into my gradient?
>
> It would be ideal if we could see some actual effect of gradient flow into the policy due to the knowledge of the action mask in any one of the methods and experimental results. That would resolve all my concerns. But even if that's not tractable right now, I would like to see some evidence or a thought experiment to prove that it is indeed possible!
>
> 3. I also tried to understand what the distributional mask's gradient in Eq. 17 physically implies. If we look at the terms in the gradient, the first term is the standard REINFORCE term, which has the implication that if the associated return R is high with this term, then the policy gradient increases $\log \pi_\theta(a | s)$ for that state. Similarly, it would reduce the second term here because of the negative sign, i.e., $\log \int_{A^r} \pi_\theta(a|s)da $ should go down if the return is high. But, this means that the probability density of $\pi_\theta(\cdot|s)$ should go down over $A^r$. Basically, the area under curve for the segment of $A^r$ has a lower probability density than the other areas in the action space. But, I don't see any reason why this should happen? This would increase the probability of "invalid actions" (along with the action $a$ from the first RL term). I see no reason why that would be helpful at all or why that makes sense?
>
> To sum up, I really want to believe that there is some performance benefit to considering action masking as part of policy and not environment. But, I don't have any evidence to support that yet. I hope the authors can help me understand this.
>
> If it turns out that there is no gradient flow, then I don't see any merit in the entire mathematical formulation of this paper, because all it says is that we should just consider action masking as part of environment, which is the trivial thing to do. I sincerely hope that is not the case, and by answering 1,2,3, we can get to that conclusion.

---

> > ### Author Response · Authors · 2024-08-09
> > **Thought experiments for gradient flow and non-bijectivity (1/2)**
> >
> > Thank you for your detailed response and thoughts on the intuitive and empirical effect of the changed gradient.
> >
> > **1.**
> >
> > Thank you for your illustrative example. The mapping function $\mathcal{F}$ describes the relation between the action space and the relevant action space. Thus, using  $\nabla \log \pi^r_\theta(a^r \| s)$ instead of $\nabla \log \pi_\theta(a \| s)$ in the objective of PPO encodes this mapping, i.e., is the extra information (see 2. second paragraph for a numerical example).
> >
> > Additionally, we think that the example, doesn't directly apply to our evaluation of stochastic policy gradients. In this context, the neural network's output is not the action itself but rather the parameters (such as the mean) of a probability distribution from which actions are sampled. The function in question is applied to these samples, effectively modifying the probability distribution.
> > Since the policy gradient assumes the log-likelihood to come from the distribution the action is sampled from, we argue that it is necessary to derive this gradient $\nabla_\theta \log \pi^r_\theta (a | s)$ for the modified distribution.
> > To illustrate this point, consider the scenario where the policy neural network models a beta distribution instead of a normal distribution (as explored in previous work [Chou et al., 2017, Improving stochastic ...]). In this case, would you agree that it is necessary to use the gradient of the beta distribution?
> >
> > We are not certain that we fully covered all your questions in 1. Could you please elaborate if that is not the case.
> >
> >
> > **2.**
> >
> > Thank you for your question. Intuitively, a non-bijective mapping function in the environment means that two different actions chosen by the agent can lead to exactly the same next state. Consider this abstract example. The function $g(a) = 0$ maps the samples of the policy distribution to the zero vector, making it a non-bijective function. Essentially, no matter the parameters of the neural network, the action would always be $0$, thus making any gradient updated irrelevant. If we incorporate $g(a)$ in the environment, a gradient update would still occur since the agent receives a return and $\log \pi_\theta (a | s)$ represents the log-likelihood of the unmapped sample $a$. However, if we treat $g(a)$ as part of the policy distribution, we essentially transform $\pi_\theta(a| s)$ into a Dirac distribution at $0$. Since the Dirac distribution is independent of the parameters $\theta$, the gradient $\nabla_\theta \log \pi_\theta^r(a| s)$ would be zero, which more accurately reflects the situation, as no meaningful updates to the policy can be made.
> >
> >
> >
> > Additionally, let us illustrate the gradient flow with a numerical example for the generator mask. Consider a relevant action zonotope with center $c = [0, 0]$ and generator matrix $G$:
> > $$
> >     G = \begin{bmatrix}
> > 1 & 0 & 0.5\\\\
> > 0 & 1 & 0.5
> > \end{bmatrix}.
> > $$
> >
> > The policy $\pi(a|s)$ is defined by a mean vector for the multi-variate Normal distribution $\mu_\theta = [0.4, 0.4, 0.4]$ and an identity covariance matrix. The relevant policy is defined as in Eq. 11. Now consider two samples: $a_1 = [0.5, 0.5, 0]$ and $a_2 = [0, 0, 1]$. Both actions result in the same relevant action $a^r = [0.5,0.5]$. The table below provides the numerical evaluation for the gradient at $\mu_\theta$ and the log probabilities for the two actions.
> >
> > |         | $\nabla_{\mu_\theta} \log \pi_\theta(a_i \| s)$ | $\nabla_{\mu_\theta} \log \pi_\theta^r(a^r\|s)$ | $\log\pi_\theta(a_i\|s)$ | $\log\pi_\theta^r(a^r\|s)$ |
> > |---------|------------------------------------|--------------------------------------|--------------------|--------------------|
> > | $a_1$ | [0.1, 0.1, -0.4]                   | [-0.066, -0.066, -0.066]             | -2.84              | -2.04              |
> > | $a_2$ | [-0.4, -0.4, 0.6]                  | [-0.066, -0.066, -0.066]             | -3.09              | -2.04              |
> >
> > This numerical example shows that $\pi^r$ leads to the same log-likelihood, reflecting that the executed relevant actions are identical. If we look at the gradient flow at $\mu_\theta$, we see that for $\pi^r$, the gradient is the same for both $a_1$ and $a_2$, thus providing consistent feedback. In contrast, for the gradient of $\pi$, we observe in this example two contradictory gradients, although the executed relevant action is identical. Thus, potentially introducing an oscillation effect for the parameter update.
> >
> > Let us provide a physical interpretation of this numerical example as well. The relevant action set described with $G$ and $c$ has two generators that are axis aligned in $\mathcal{A}$ and one that influences both dimensions of $\mathcal{A}$. A gradient step with respect to \pi^r$ includes this state-dependent but fixed dependency between the actions. If the generator mask mapping function is part of the environment and, thus, a black box for the agent, it has to learn this dependency with potentially many training samples.
> >
> > (1/2)

---

> > > ### Author Response · Authors · 2024-08-09
> > > **Physical implications of the gradient for the distributional mask (2/2)**
> > >
> > > **3.**
> > >
> > > Thank you for this question; it is indeed a counterintuitive observation. You are correct that the policy gradient aims to increase the log-likelihood of high-return samples by adapting the parameters of the Gaussian policy distribution. However, the situation becomes more complex with the distributional mask, as we explain in the following. The relevant policy distribution $\pi_\theta^r(a^r| s)$ essentially models a truncated normal distribution in one dimension [Burkardt, 2023, The Truncated Normal Distribution]. Due to the normalization of the distribution with the integral over the relevant action zonotope (Eq. 15), there are now two ways to increase the log-likelihood of a sample.
> > > 1. Moving the mean closer to the sample (as reflected in the first term of Eq. 17).
> > > 2. Moving the mean farther away decreases the integral in Eq. 15, thereby increasing the log-likelihood (as reflected in the second term of Eq. 17).
> > >
> > > For the second option, while the probability density of $\pi_\theta(\cdot| s)$ decreases over $\mathcal{A}^r$, it remains the same for $\pi_\theta^r(\cdot| s)$ due to the normalization. These effects would be easier to grasp with a visualization of the distributions, which we cannot provide due to the format restrictions of comments. So please let us know if we should elaborate more.
> > >
> > >
> > > Finally, we are curious to understand better why you consider action masking for convex sets in the environment as trivial? While it may simplify some aspects by excluding the gradient adaption, the core effort still lies in the formulation of the three masking approaches, which we believe is a key contribution of our work.
> > >
> > > We hope our answers resolve your skepticism regarding the relevance of adding the mapping function to the policy. Please let us know if you have additional questions. Lastly, let us take a step back from the theoretical analysis and look at the discussion from a practical point of view. As we stated in the summary rebuttal, defining action masking in the environment completely excludes the distributional mask and practically excludes the generator mask. This is because we would need to modify the environment when changing the relevant action zonotope's shape, leading to impracticalities like multiple versions of the Walker2D environment with varying numbers of generators.
> > > In summary, formulating our masking approaches as part of the policy reflects the actual policy distribution, is more intuitive and practical, and allows for more formulations of $\mathcal{F}$.
> > >
> > > (2/2)

---

> > > ### Comment · Reviewer_8567 · 2024-08-11
> > > **Followup**
> > >
> > > 1. Your analogy about beta distribution is not correct. If your model was to output a normal distribution $\mu_\theta$ and $\sigma_\theta$ --> then you sample from it --> then you use that action to somehow create your beta distribution; then does the gradient flowing into mu, sigma differ whether or not you consider the presence of beta distribution in the latter part?
> > >
> > > Your paper formulates g as a function that applies over the sampled action, right? $a^r = g(a) = c + G a$
> > >
> > > 2. The real question is not whether $\nabla_{\mu_\theta} \log \pi_\theta(a_i \mid s)$ and $\nabla_{\mu_\theta} \log \pi_\theta^r(a^r \mid s)$ are different. Obviously, they will be different. The question is that, in the backward pass of your update step, would the gradient flowing into $\mu_\theta$ and $\Sigma_\theta$ change at all due to the presence of G? Wouldn't the backpropagation remove the effect of G because of chain rule? In that case, the presence of the transformation g, does not add anything to the policy gradient flowing into $\theta$, which are the parameters of the neural network.
> > >
> > > Are you claiming that if I train two policies:
> > > 1. With $\pi_\theta$, do a transformation $a^r = g(a) = c + G a$ in the environment, get rewards and update the $\theta$,
> > > 2. With $\pi^r_\theta$ as defined in Equation (11), get rewards and update the $\theta$,
> > > then there would actually be a difference in what update happens to $\theta$ ?
> > >
> > >
> > > I looked into your code and I don't see any evidence of modification of the loss function. All I see are different ways of applying the mask with your policy network. Then, how why do you expect the policy gradient flowing into $\theta$ change at all because of a presence of a post-processing function g?

---

> > > > ### Author Response · Authors · 2024-08-11
> > > > **Answer to followup (1/2)**
> > > >
> > > > Thank you for your follow-up and in-depth engagement with our work, which we really appreciate.
> > > >
> > > > > Your paper formulates g as a function that applies over the sampled action, right? $a^r = g(a) = c + G a$
> > > >
> > > > For the generator mask, yes, resulting in samples from $\pi_\theta^r$.
> > > >
> > > > > The question is that, in the backward pass of your update step, would the gradient flowing into $\mu_\theta$ and $\Sigma_\theta$ change at all due to the presence of G? Wouldn't the backpropagation remove the effect of G because of chain rule?
> > > >
> > > > Thanks for your question. Since the loss function does not use the sampled actions directly, the effect of action masking is not simply the derivative of $G a + c$, i.e., modelled with the chain rule. Instead, the sampled actions are integrated into the objective of PPO through the policy probability distribution.
> > > > We change $\log \pi(a|s)$ to $\log \pi^r(a^r|s)$ where $\pi^r(a^r|s)$ is defined as
> > > > $$
> > > > \pi_\theta^r (a^r|s) = \mathcal{N}(a^r; G \mu_\theta + c, G \Sigma_\theta G^T).
> > > > $$
> > > > Note that this is Eq. 11 with corrected notation, where we can imagine confusion could originate.
> > > > In Proposition 3, we derive the changes of the gradient due to the change of policy distribution to $\pi_\theta^r (a^r|s)$. Let us have a closer look at Eq. 13 when replacing $a^r$ with $Ga +c$:
> > > > $$
> > > > \nabla_{\mu_\theta} \log \pi_\theta^r(a^r | s)
> > > >     =  G^T (G \Sigma_\theta G^T)^{-1} (a^r - c - G \mu_\theta) = G^T (G \Sigma_\theta G^T)^{-1} (c + G a - c - G \mu_\theta) =
> > > >     G^T (G \Sigma_\theta G^T)^{-1} (G a - G \mu_\theta).
> > > > $$
> > > > From the transformed Eq. 13, we can observe that the effect of $G$ does not cancel out. We would like to refer you to appendix A.2 for the detailed proof.
> > > >
> > > > Let us also have another look at the numerical example to clarify it further. In this example, we calculate the first column with the loss $\log \pi(a_i | s)$ and the second with the loss $\log \pi^r(a^r | s)$. These loss functions reflect the policy gradient loss with assumed advantage $A^\pi=1$ for a single action sample, i.e., $a_1$ or $a_2$. While this is not the entire PPO loss, it reflects the only adaption we make: changing the policy distribution to $\pi^r$. Thus, the first two columns provide a simplified numerical example for the backward pass, i.e., gradient flow. For further clarity, assume that $\mu_\theta = \theta$, then for $a_1$ the parameters $\theta$ would be updated with $\lambda [0.1, 0.1, -0.4]$ and $\lambda [-0.066, -0.066, -0.066]$ when calculated without and with the correction of the policy distribution, respectively. Note that $\lambda$ is the learning rate.
> > > >
> > > >
> > > > > Are you claiming that if I train two policies: ... there would actually be a difference in what update happens to $\theta$?
> > > >
> > > > Yes. For a numerical example, see the above answer.
> > > >
> > > >
> > > > > I looked into your code and I don't see any evidence of modification of the loss function. All I see are different ways of applying the mask with your policy network. Then, how why do you expect the policy gradient flowing into $\theta$ change at all because of a presence of a post-processing function g?
> > > >
> > > > Thank you for looking into our code. You are correct in observing that we do not directly modify the loss function. However, our action masking framework also does not require such a modification. The change in gradient comes from how the log-likelihood of the sampled action is computed. Standard PPO uses the normal distribution for $\pi$, whose mean is defined by the output of the neural network policy, to compute the log-likelihood. We define the modified normal distribution $\pi^r$ as in Eq. 11 (with corrected notation), which is implemented in lines 41-43 in `action_masking/rlsampling/sb3_distributions/generator_dist.py`. This modified distribution is used to compute the `log_prob` in line 535 of `action_masking/sb3_contrib/ppo/ppo.py`, which in terms is used to compute the loss function. The automatic differentiation engine of PyTorch now computes the accurate gradients $\nabla_\theta \log \pi_\theta^r(a | s)$, as derived in Proposition 3 (Eq. 13 and 14). We have implemented Eq. 13 and 14 and compared them to the PyTorch's gradients to validate their correctness.
> > > >
> > > > In essence, the use of our modified distribution in the log-likelihood calculation leads to the desired change in gradients, as defined by our theoretical analysis. We see that our code is not easy to navigate with respect to this implementation change and will clarify this in the README.md.

---

> > > > > ### Author Response · Authors · 2024-08-11
> > > > > **Answer to followup (2/2)**
> > > > >
> > > > > > Your analogy about beta distribution is not correct. If your model was to output a normal distribution $\mu_\theta$ and $\sigma_\theta$ --> then you sample from it --> then you use that action to somehow create your beta distribution; then does the gradient flowing into mu, sigma differ whether or not you consider the presence of beta distribution in the latter part?
> > > > >
> > > > > We appreciate your concern about the beta distribution analogy, and we acknowledge that it is somewhat abstract. In the following, we want to show why we still think it is correct and also applies to our formulation.
> > > > >
> > > > > Let us start by a short introduction on implementations of univariate distributions. Here, the inverse transform sampling is used to sample from any univariate distribution with probability density function $\mathbb{P}(a; \theta)$ that has an invertible cumulative density function (CDF) $F(a; \theta)$. The method takes a uniform sample $a \sim U(0, 1)$, and applies the inverse cumulative density function (ICDF) $F^{-1}(a; \theta)$ to create samples from the distribution $\mathbb{P}(a; \theta)$.
> > > > >
> > > > > For our example, let $x,y \in \mathbb{R}^N$ be two outputs of a neural network. We treat them as the two parameters of a normal distribution by sampling $a \sim \mathbb{P}(a; x, y) = \mathcal{N}(x,y)$, i.e. $\mu = x$, $\sigma = y$. If we use the gradients $\nabla_x$ and $\nabla_y$ of the log-likelihood of the normal distribution in the backward pass, $x$ and $y$ will be shaped to represent the mean of variance of the normal distribution.
> > > > > Now, instead, we apply the function $g(a) = F^{-1}\_{Beta} ( F\_{Normal} (a; x, y); x, y )$, which first maps the samples of the normal distribution to the uniform distribution $U(0, 1)$, and then transforms them to samples of the beta distribution through the ICDF. Thereby we **exactly** create samples from the beta distribution parameterized with $\alpha=x$ and $\beta=y$, i.e. $a \sim \mathbb{P}(a; x, y) = \frac{a^{x-1} (1 - a)^{y-1})}{B(x, y)}$, where $B(x,y)$ is the beta function.
> > > > > If we now still use the gradients of the log-likelihood of the normal distribution to update $x$ and $y$, we shape them to represent $\mu$ and $\sigma$ instead of $\alpha$ and $\beta$, which would result in $\mathbb{P}(a; x, y)$ representing beta distributions with likely not meaningful shapes. Therefore, we should use the gradients $\nabla_x$ and $\nabla_y$ of the log-likelihood of the beta distribution to shape $x$ and $y$ to meaningful values of $\alpha$ and $\beta$.
> > > > >
> > > > > We want to reiterate that this example is very abstract, since there is a much more straightforward way to sample from the beta distribution. However, it indicates that it is important to use the gradients of the distribution that actually produce the samples in order to learn meaningful values for the parameters of the distribution.

---

> > > > > > ### Comment · Reviewer_PubC · 2024-08-12
> > > > > > **Thoughts regarding action masking as part of the policy versus the environment**
> > > > > >
> > > > > > It's interesting to follow along with this detailed discussion of the effects of masking in the policy, I think the core issue at hand (at least to me) is whether there's major benefits to treating action masking as part of the policy, which complicates the math (as noted by reviewer 8567) and limits to only differentiable masks. The author's arguments regarding interpretability, learning efficiency, and allowing more types of masks are valid (though I don't see theoretical or experimental evidence for the learning efficiency argument, I buy that it should intuitively provide some -possibly small- benefit), but the degree to which those factors matter is unclear, and as such there's not a decisive answer to the question of policy-masking versus environment-masking.
> > > > > >
> > > > > > It seems to me that ultimately there's a tradeoff to be made, and it's not clear from existing evidence what the better tradeoff to make is. That's okay for this paper- this work focuses on one approach, and while that approach has advantages and disadvantages it's not necessary for one paper to compare to other, not previously described, approaches to validate the usefulness of the one the authors favor.
> > > > > >
> > > > > > That said, I do think this concern is coming up because there's an obvious (and much simpler) alternative formulation which could be compared against. I do think it would greatly strengthen the paper (and the contribution of this research direction as a whole) to experimentally compare policy-masking versus environment-masking to confirm that the supposed advantages of policy-masking actually occur compared to the environment-masking approach. At the least I'd encourage the authors to investigate such comparisons in future work regardless of the decision for this submission.

---

> > > > > > > ### Author Response · Authors · 2024-08-12
> > > > > > > **Answer to "Thoughts regarding action masking as part of the policy versus the environment"**
> > > > > > >
> > > > > > > Thank you sincerely for this comment as it identifies the core issue of the discussion: Whether to treat masking as part of the policy brings major benefit compared to treating it as part of the environment. Still we want to address some points in the following.
> > > > > > >
> > > > > > > > ... which complicates the math (as noted by reviewer 8567) and limits to only differentiable masks.
> > > > > > >
> > > > > > > We don't think that our approach limits to only differentiable masks. The ray mask is not always differentiable, since computing $g(a)$ for it can require solving optimization problems to compute the boundary points of certain set representations (e.g. zonotopes). We can still include it into our action masking framework, because we prove in Proposition 1 that the gradient remains unchanged compared to the original normal distribution.
> > > > > > >
> > > > > > > > The author's arguments regarding interpretability, learning efficiency, and allowing more types of masks are valid (though I don't see theoretical or experimental evidence for the learning efficiency argument, ...
> > > > > > >
> > > > > > > Your assessment for the lack of practical evidence is valid. Still, is the derivation of the actual gradients of the changed distributions in the paper, and the numerical example we provided in previous answers, not at least theoretical evidence for the learning efficiency argument?
> > > > > > >
> > > > > > > > It seems to me that ultimately there's a tradeoff to be made, and it's not clear from existing evidence what the better tradeoff to make is. That's okay for this paper- this work focuses on one approach, and while that approach has advantages and disadvantages it's not necessary for one paper to compare to other, not previously described, approaches to validate the usefulness of the one the authors favor.
> > > > > > >
> > > > > > > Thank you for this favourable comment!
> > > > > > >
> > > > > > > > That said, I do think this concern is coming up because there's an obvious (and much simpler) alternative formulation which could be compared against.
> > > > > > >
> > > > > > > We agree that treating action masking as part of the environment would be simpler, but is it actually much simpler? Sure, the gradient derivation could be omitted, but the main effort lies in the implementation of the actual masking procedure, which remains in its entirety. Implementing the adapted gradient for e.g. the generator mask adds just a few additional lines of code with automatic differentiation engines like PyTorch (as we have detailed in the previous answer).
> > > > > > >
> > > > > > > > I do think it would greatly strengthen the paper (and the contribution of this research direction as a whole) to experimentally compare policy-masking versus environment-masking to confirm that the supposed advantages of policy-masking actually occur compared to the environment-masking approach. At the least I'd encourage the authors to investigate such comparisons in future work regardless of the decision for this submission.
> > > > > > >
> > > > > > > We fully agree that an experimental comparison would strengthen the paper. As we argued in previous answers, we do already provide this comparison with the ray mask. Since its gradient does not need to be adapted, our formulation is mathematically equivalent to viewing the ray mask as part of the environment. Also the only implementation change that would need to be made is moving the line `masked_action = ray_mask(action)` from the policy class to a few lines later in the `environment.step()` method, which does not change the inner workings of the code. As for the generator mask, we previously argued that it would be highly impractical to implement it into the environment as it leads to a change in the environment when changing the shape of the relevant action set (see answer "Physical implications of the gradient for the distributional mask (2/2)").
> > > > > > > In summary, **we do already provide the requested comparison, and the results are not in favour of environment masking**. We just do not explicitly state it in the paper, which we can of course change. However, then we would have to introduce the concept of environment masking, which is not an established concept, and was not formulated prior to this discussion round.

---

### Decision · Program_Chairs · 2024-09-25

**Decision:**

Accept (poster)

**Comment:**

The paper proposes three continuous action masking methods to focus reinforcement learning on state-specific relevant actions, enhancing training efficiency, effectiveness, and safety. Although there is consensus about the overall positive evaluation of this paper, the reviewers suggest that the paper would benefit from (a) additional experiments supporting the need for the gradient derivations. (b) improved clarity of presentation. Apart from this, the reviews indicate that evaluations on more complex domains would strengthen the work.